# QUASI-MONTE CARLO FOR 3D SLICED WASSERSTEIN

**Khai Nguyen, Nicola Bariletto & Nhat Ho**
Department of Statistics and Data Sciences
The University of Texas at Austin
Austin, TX 78712, USA
{khainb,nicola.bariletto,minhnhat}@utexas.edu

## ABSTRACT

Monte Carlo (MC) integration has been employed as the standard approximation method for the Sliced Wasserstein (SW) distance, whose analytical expression involves an intractable expectation. However, MC integration is not optimal in terms of absolute approximation error. To provide a better class of empirical SW, we propose quasi-sliced Wasserstein (QSW) approximations that rely on Quasi-Monte Carlo (QMC) methods. For a comprehensive investigation of QMC for SW, we focus on the 3D setting, specifically computing the SW between probability measures in three dimensions. In greater detail, we empirically evaluate various methods to construct QMC point sets on the 3D unit-hypersphere, including the Gaussian-based and equal area mappings, generalized spiral points, and optimizing discrepancy energies. Furthermore, to obtain an unbiased estimator for stochastic optimization, we extend QSW to Randomized Quasi-Sliced Wasserstein (RQSW) by introducing randomness in the discussed point sets. Theoretically, we prove the asymptotic convergence of QSW and the unbiasedness of RQSW. Finally, we conduct experiments on various 3D tasks, such as point-cloud comparison, point-cloud interpolation, image style transfer, and training deep point-cloud autoencoders, to demonstrate the favorable performance of the proposed QSW and RQSW variants[1].

## 1 INTRODUCTION

The Wasserstein (or Earth Mover's) distance (Peyré & Cuturi, 2020) has been widely recognized as a geometrically meaningful metric for comparing probability measures. For instance, it has been successfully employed in various applications such as generative modeling (Salimans et al., 2018), domain adaptation (Courty et al., 2017), clustering (Ho et al., 2017), and so on. Specifically, the Wasserstein distance serves as the standard metric for applications involving 3D data, such as point-cloud reconstruction (Achlioptas et al., 2018), point-cloud registration (Shen et al., 2021), point-cloud completion (Huang et al., 2023), point-cloud generation (Kim et al., 2020), mesh deformation (Feydy et al., 2017), image style transfer (Amos et al., 2023), and various other tasks.

Despite its appealing features, the Wasserstein distance exhibits high computational complexity. When using conventional linear programming solvers, evaluating the Wasserstein distance carries a $\mathcal{O}(n^3 \log n)$ time complexity (Peyré & Cuturi, 2020), particularly when dealing with discrete probability measures supported on at most $n$ atoms. Furthermore, computing the Wasserstein distance has at least $\mathcal{O}(n^2)$ space complexity, which is related to storing the pairwise transportation cost matrix. The Sliced Wasserstein (SW) distance (Bonneel et al., 2015) stands as a rapid alternative metric to the plain Wasserstein distance. Since the SW distance is defined as a sliced probability metric based on the Wasserstein distance, it is equivalent to the latter while enjoying appealing properties (Nadjahi et al., 2020). More importantly, the time complexity and space complexity of the SW metric are only $\mathcal{O}(n \log n)$ and $\mathcal{O}(n)$, respectively. As a result, the SW distance has been successfully adopted in various applications, including domain adaptation (Lee et al., 2019), generative models (Nguyen & Ho, 2024; Nguyen et al., 2024), clustering (Kolouri et al., 2018), gradient flows (Bonet et al., 2022), Bayesian inference (Yi & Liu, 2021), and more. In the context of 3D data analysis, the SW distance is employed in numerous applications such as point-cloud registration (Lai & Zhao, 2017),

---

[1]Code for the paper is published at https://github.com/khainb/Quasi-SW.

reconstruction, and generation (Nguyen et al., 2023), mesh deformation (Le et al., 2024a), shape matching (Le et al., 2024b), image style transfer (Li et al., 2022), along with various other tasks.

Formally, the SW distance is defined as the expectation of the Wasserstein distance between two one-dimensional projected measures under the uniform distribution over projecting directions, i.e., the unit hypersphere. Exact computation of the SW distance is well-known to be intractable; hence, in practice, it is estimated empirically through Monte Carlo (MC) integration. Specifically, (pseudo-)random samples are drawn from the uniform distribution over the unit hypersphere to approximate the analytical integral. However, the approximation error of MC integration is suboptimal because (pseudo-)uniform random samples may not exhibit sufficient "uniformity" over the space (Owen, 2013). Quasi-Monte Carlo (QMC) methods (Keller, 1995) address this issue by building deterministic point sets, known as "low-discrepancy sequences", on which to evaluate the integrand. Low discrepancy implies that the points are more "uniform" and provide a superior approximation of the uniform expectation over the domain, compared to randomly drawn points.

Conventional QMC methods primarily focus on integration over the unit hypercube $[0, 1]^d$ (for $d \geq 1$). To assess the uniformity of a point set on $[0, 1]^d$, a widely employed metric is the "star-discrepancy" (Koksma, 1942). A lower star-discrepancy value typically results in reduced approximation error, as per the Koksma–Hlawka inequality (Koksma, 1942). When a point set exhibits a sufficiently small star-discrepancy, it is referred to as a "low-discrepancy sequence". For the unit cube, several options exist, such as the Halton sequence (Halton & Smith, 1964), the Hammersley point set (Hammersley, 2013), the Faure sequence (Faure, 1982), the Niederreiter sequence (Niederreiter, 1992), and the widely used Sobol sequence (Sobol, 1967). QMC integration is renowned for its efficiency and effectiveness, especially in low (e.g., 3) dimensions.

**Contribution.** In short, we integrate QMC methodologies into the framework for SW distance computation. Specifically, our contributions are three-fold:

1. As the SW distance involves integration over the unit hypersphere of dimension $d - 1$, rather than the well-studied (for QMC purposes) hypercube, we provide an overview of practical methods for constructing point sets on the unit hypersphere, which can serve as candidates for low-discrepancy sequences (referred to as QMC point sets). Specifically, our exploration encompasses the following techniques: (i) mapping a low-discrepancy sequence from the 3D unit cube to the unit sphere using the normalized inverse Gaussian CDF, (ii) transforming a low-discrepancy sequence from the 2D unit grid to the unit sphere via the Lambert equal-area mapping, (iii) using generalized spiral points, (iv) maximizing pairwise absolute discrepancy, (v) minimizing the Coulomb energy. Notably, we believe that our work is the first to make use of the recent numerical formulation of spherical cap discrepancy (Heitsch & Henrion, 2021) to assess the uniformity of the aforementioned point sets.

2. We introduce the family of Quasi-Sliced Wasserstein (QSW) deterministic approximations to the SW distance, based on QMC point sets. Furthermore, we establish the asymptotic convergence of QSW to the SW distance, as the size of the point set grows to infinity, for nearly all constructions of QMC point sets. For stochastic optimization, we present Randomized Quasi-Monte Carlo (RMQC) methods applied to the unit sphere, resulting in Randomized Quasi-Sliced Wasserstein (RQSW) estimations. In particular, we explore two approaches for generating random point sets on $\mathbb{S}^{d-1}$: transforming randomized point sets from the unit cube and random rotation. We prove that nearly all variants of RQSW provide unbiased estimates of the SW distance.

3. We empirically demonstrate that QSW and RQSW offer better approximations of the SW distance in 3D applications. Specifically, we first establish that QSW provides a superior approximation to the population SW distance compared to conventional Monte Carlo (MC) approximations when comparing 3D empirical measures over point clouds. Then, we conduct experiments involving point-cloud interpolation, image style transfer, and training deep point-cloud autoencoders to showcase the superior performance of various QSW and RQSW variants.

**Organization.** The remainder of the paper is organized as follows. We first provide some background on the SW distance, MC estimation, and QMC methods in Section 2. Then, we discuss how to construct QMC point sets on $\mathbb{S}^{d-1}$, define QSW and RQSW approximations, and discuss some of their theoretical properties in Section 3. Section 4 contains experiments on point-cloud autoencoders, image style transfer, and deep point-cloud reconstruction. We conclude the paper in Section 5. Finally, we defer the proofs of key results, related work, and additional material to the Appendices.

**Notation.** For any $d \geq 2$, we define the unit hypersphere $\mathbb{S}^{d-1} := \{\theta \in \mathbb{R}^d \mid ||\theta||_2^2 = 1\}$, and denote the uniform distribution on it as $\mathcal{U}(\mathbb{S}^{d-1})$. For $p \geq 1$, $\mathcal{P}_p(\mathcal{X})$ represents the set of all probability measures on the set $\mathcal{X}$ that have finite $p$-moments. We denote $\theta\sharp\mu$ as the push-forward measure $\mu \circ f_\theta^{-1}$ of $\mu$ through the function $f_\theta : \mathbb{R}^d \to \mathbb{R}$ defined as $f_\theta(x) = \theta^\top x$. For a vector $X = (x_1, \ldots, x_m) \in \mathbb{R}^m$, $P_X$ represents the empirical measure $\frac{1}{m}\sum_{i=1}^m \delta_{x_i}$.

## 2 BACKGROUND

In Section 2.1, we define the SW distance and review the standard MC approach to estimate it. After that, in Section 2.2, we delve into QMC methods for approximating integrals over the unit hypercube.

### 2.1 SLICED WASSERSTEIN DISTANCE AND MONTE CARLO ESTIMATION

**Definitions.** Given $p \geq 1$, the Sliced Wasserstein (SW) distance of order $p$ (Bonneel et al., 2015) between two probability measures $\mu, \nu \in \mathcal{P}_p(\mathbb{R}^d)$ (i.e., with finite $p^{th}$ moment) is defined as

$$\mathrm{SW}_p^p(\mu, \nu) := \mathbb{E}_{\theta\sim\mathcal{U}(\mathbb{S}^{d-1})}[\mathrm{W}_p^p(\theta\sharp\mu, \theta\sharp\nu)], \tag{1}$$

where $\mathrm{W}_p(\theta\sharp\mu, \theta\sharp\nu)$ is the one-dimensional Wasserstein between the projections of $\mu$ and $\nu$ along direction $\theta$. As mentioned, one has the closed-form $\mathrm{W}_p^p(\theta\sharp\mu, \theta\sharp\nu) = \int_0^1 |F_{\theta\sharp\mu}^{-1}(z) - F_{\theta\sharp\nu}^{-1}(z)|^p dz$, where $F_{\theta\sharp\mu}^{-1}(\cdot)$ and $F_{\theta\sharp\nu}^{-1}(\cdot)$ are the inverse cumulative distribution functions of $\theta\sharp\mu$ and $\theta\sharp\nu$.

**Monte Carlo estimation.** To approximate the intractable expectation in the SW distance formula, MC samples are generated and give rise to the following estimate:

$$\widehat{\mathrm{SW}}_p^p(\mu, \nu; L) = \frac{1}{L}\sum_{l=1}^L \mathrm{W}_p^p(\theta_l\sharp\mu, \theta_l\sharp\nu), \tag{2}$$

where random samples $\theta_1, \ldots, \theta_L$ (referred to as *projecting directions*) are drawn i.i.d. from $\mathcal{U}(\mathbb{S}^{d-1})$. When $\mu$ and $\nu$ are discrete probability measures that have at most $n$ supports, the time complexity of to compute $\widehat{\mathrm{SW}}_p$ is $\mathcal{O}(Ln\log n + Ldn)$, while the corresponding space complexity is $\mathcal{O}(Ld + Ln)$. We refer to Algorithm 1 in Appendix B for more details on the computation of (2).

**Monte Carlo error.** Similar to other usages of MC, the approximation error of the SW decreases at $\mathcal{O}(L^{-1/2})$ rate. In greater detail, a general upper-bound (Nadjahi et al., 2020) is:

$$\mathbb{E}_{\theta_1,\ldots,\theta_L\overset{\text{iid}}{\sim}\mathcal{U}(\mathbb{S}^{d-1})}\left[|\widehat{\mathrm{SW}}_p^p(\mu, \nu; L) - \mathrm{SW}_p^p(\mu, \nu)|\right] \leq \frac{1}{\sqrt{L}}\mathrm{Var}_{\theta\sim\mathcal{U}(\mathbb{S}^{d-1})}\left[\mathrm{W}_p^p(\theta\sharp\mu, \theta\sharp\nu)\right]^{1/2}.$$

### 2.2 QUASI-MONTE CARLO METHODS

**Problem.** Conventional Quasi-Monte Carlo (QMC) methods focus on approximating an integral $I = \int_{[0,1]^d} f(x)dx = \mathbb{E}_{x\sim\mathcal{U}([0,1]^d)}[f(x)]$ on the unit hypercube $[0, 1]^d$, with $\mathcal{U}([0, 1]^d)$ denoting the corresponding uniform distribution. Similarly to MC methods, QMC integration also approximates the expectation with an equal weight average $\hat{I}(L) = \frac{1}{L}\sum_{l=1}^L f(x_l)$. However, the point set $\theta_1, \ldots, \theta_L$ is constructed differently.

**Low-discrepancy sequences.** QMC requires a point set $x_1, \ldots, x_L$ such that $\hat{I}(L) \to I$ as $L \to \infty$, and aims to obtain high uniformity. To measure the latter, the star discrepancy (Owen, 2013) has been used: $D^*(x_1, \ldots, x_L) = \sup_{x\in[0,1)^d} |F_L(x|x_1, \ldots, x_L) - F_{\mathcal{U}([0,1]^d)}(x)|$, where $F_L(x|x_1, \ldots, x_L) = \frac{1}{L}\sum_{l=1}^L \mathbf{1}_{x_l\leq x}$ (the empirical CDF) and $F_{\mathcal{U}([0,1]^d)}(x) = \mathrm{Vol}([0, x])$ is the CDF of the uniform distribution over the unit hypercube. Since the star discrepancy is the sup-norm between the empirical CDF and the CDF of the uniform distribution, the points $x_1, \ldots, x_L$ are asymptotically uniformly distributed if $D^*(x_1, \ldots, x_L) \to 0$. Moreover, there is a connection between the star discrepancy and the approximation error (Hlawka, 1961) via the Koksma-Hlawka inequality. In particular, we have:

$$|\hat{I}(L) - I| \leq D^*(x_1, \ldots, x_L)\mathrm{Var}_{HK}(f), \tag{3}$$

where $\text{Var}_{HK}(f)$ is the total variation of $f$ in the sense of Hardy and Krause (Niederreiter, 1992). Formally, $x_1, \ldots, x_L$ is called a *low-discrepancy sequence* if $D^*(x_1, \ldots, x_L) \in \mathcal{O}(L^{-1}\log(L)^d)$. Therefore, QMC integration can achieve better approximation than its MC counterpart if $L \geq 2^d$, since the error rate of MC is $\mathcal{O}(L^{-1/2})$. In relatively low dimensions, e.g., three dimensions, QMC gives a better approximation than MC. Several such sequences have been proposed, e.g., the Halton sequence (Halton & Smith, 1964), the Hammersley point set (Hammersley, 2013), the Faure sequence (Faure, 1982), the Niederreiter sequence Niederreiter (1992), and the Sobol sequence (Sobol, 1967). We refer the reader to Appendix B for the construction of the Sobol sequence.

# 3 QUASI-MONTE CARLO FOR 3D SLICED WASSERSTEIN

In Section 3.1, we explore the construction of candidate point sets as low-discrepancy sequences on the unit hypersphere. Subsequently, we introduce Quasi-Sliced Wasserstein (QSW), Randomized Quasi-Sliced Wasserstein (RQSW) distance, and discuss their properties in Section 3.2-3.3.

## 3.1 LOW-DISCREPANCY SEQUENCES ON THE UNIT-HYPERSPHERE

**Spherical cap discrepancy.** The most used discrepancy to measure the uniformity of a point set $\theta_1, \ldots, \theta_L \in \mathbb{S}^{d-1}$ is the spherical cap discrepancy (Brauchart & Dick, 2012):

$$D^*_{\mathbb{S}^{d-1}}(\theta_1, \ldots, \theta_L) = \sup_{w \in \mathbb{S}^{d-1}, t \in [-1,1]} \left| \frac{1}{L} \sum_{l=1}^{L} \mathbf{1}_{\theta_L \in C(w,t)} - \sigma_0(C(w,t)) \right|, \tag{4}$$

where $C(w,t) = \{x \in \mathbb{S}^{d-1} | \langle w, x \rangle \leq t\}$ is a spherical cap, and $\sigma_0$ is the law of $\mathcal{U}(\mathbb{S}^{d-1})$. It is proven that $\theta_1, \ldots, \theta_L$ are asymptotically uniformly distributed if $D^*_{\mathbb{S}^{d-1}}(\theta_1, \ldots, \theta_L) \to 0$ (Brauchart & Dick, 2012). A point set $\theta_1, \ldots, \theta_L$ is called a low-discrepancy sequence on $\mathbb{S}^2$ if $D^*_{\mathbb{S}^2}(\theta_1, \ldots, \theta_L) \in \mathcal{O}(L^{-3/4}\sqrt{\log(L)})$. For some functions belonging to suitable Sobolev spaces, a lower spherical cap discrepancy leads to a better worse-case error (Brauchart & Dick, 2012; Brauchart et al., 2014).

**QMC point sets on $\mathbb{S}^{d-1}$.** We explore various methods to construct potentially low-discrepancy sequences on the unit hypersphere. Some of these constructions are applicable to any dimension, while others are specifically designed for the 2-dimensional sphere $\mathbb{S}^2 \subset \mathbb{R}^3$.

*Gaussian-based mapping.* Utilizing the connection between Gaussian distribution and the uniform distribution over the unit hypersphere, i.e., $x \sim \mathcal{N}(0, I_d)$ then $x/\|x\|_2 \sim \mathcal{U}(\mathbb{S}^{d-1})$, we can map a low-discrepancy sequence $x_1, \ldots, x_L$ on $[0,1]^d$ to a potentially low-discrepancy sequence $\theta_1, \ldots, \theta_L$ on $\mathbb{S}^{d-1}$ through the mapping $\theta = f(x) = \Phi^{-1}(x)/\|\Phi^{-1}(x)\|_2$, where $\Phi^{-1}$ is the inverse CDF of $\mathcal{N}(0,1)$ (entry-wise). This technique is mentioned in (Basu, 2016) and can be used in any dimension.

*Equal area mapping.* Following the same idea of transforming a low-discrepancy sequence on the unit grid, we can utilize an equal area mapping (projection) to map from $[0,1]^2$ to $\mathbb{S}^2$. For instance, we use the Lambert cylindrical mapping $f(x,y) = (2\sqrt{y-y^2}\cos(2\pi x), 2\sqrt{y-y^2}\sin(2\pi x), 1-2y)$. This approach generates an asymptotically uniform sequence which is empirically shown to be low-discrepancy on $\mathbb{S}^2$ (Aistleitner et al., 2012).

*Generalized Spiral.* We can explicitly construct a set of $L$ points that are equally distributed on $\mathbb{S}^2$ with spherical coordinates $(\phi_1, \phi_2)$ (Rakhmanov et al., 1994): $z_i = 1 - \frac{2i-1}{L}, \phi_{i1} = \cos^{-1}(z_i), \phi_{i2} = 1.8\sqrt{L}\phi_{1i} \mod 2\pi$ for $i = 1, \ldots, L$. We can then retrieve Euclidean coordinates through the mapping $(\phi_1, \phi_2) \mapsto (\sin(\phi_1)\cos(\phi_2), \sin(\phi_1)\sin(\phi_2), \cos(\phi_1))$. This construction outputs an asymptotically uniform sequence (Hardin et al., 2016) which is empirically shown to achieve optimal worst-case integration error Brauchart et al. (2014) for properly defined Sobolev integrands.

*Maximizing Distance and minimizing Coulomb energy.* Previous work (Brauchart et al., 2014; Hardin et al., 2016) suggests that choosing a point set $\theta_1, \ldots, \theta_L$ which maximizes the distance $\sum_{i=1}^{L}\sum_{j=1}^{L}|\theta_i - \theta_j|$ or minimizes the Coulomb energy $\sum_{i=1}^{L}\sum_{j=1}^{L}\frac{1}{|\theta_i - \theta_j|}$ could create a potentially low-discrepancy sequence. Such sequences are also shown to achieve optimal worst-case error by Brauchart et al. (2014), though they might suffer from sub-optimal optimization in practice. Also, minimizing the Coulomb energy is proven to create an asymptotically uniform sequence (Götz, 2000). In this work, we use generalized spiral points as initialization points for optimization.

**Empirical comparison.** We adopt a recent numerical approximation for the spherical cap discrepancy (Heitsch & Henrion, 2021) to compare the discussed $L$-point sets. We visualize these sets and the corresponding discrepancies for $L = 10, 50, 100$ in Figure 6 in Appendix D.1. Overall, generalized spiral points and optimization-based points yield the lowest discrepancies, followed by equal area mapping construction. The Gaussian-based mapping construction performs worst among QMC methods; however, it still yields much lower spherical cap discrepancies than conventional random points. Qualitatively, we observe that the spherical cap discrepancy is consistent with the uniformity of point sets. We also include a comparison with the theoretical line $CL^{-3/4}\sqrt{\log(L)}$ for some constant $C$, in Figure 7 in Appendix D.1. In this case, we observe that the equal area mapping sequences, generalized spiral sequences, and optimization-based sequences seem to attain low-discrepancy, as per definition. For convenience, we refer to these sequences as QMC point sets.

## 3.2 QUASI-SLICED WASSERSTEIN

**Quasi-Monte Carlo methods for SW distances.** Based on the aforementioned QMC point sets in Section 3.1, we can define the the QMC approximation of the SW distance as follows.

**Definition 1.** *Given $p \geq 1$, $d \geq 2$, two probability measures $\mu, \nu \in \mathcal{P}_p(\mathbb{R}^d)$, and a QMC point set $\theta_1, \ldots, \theta_L \in \mathbb{S}^{d-1}$, Quasi-Sliced Wasserstein (QSW) approximation of order $p$ between $\mu$ and $\nu$ is:*

$$\widehat{QSW}_p^p(\mu, \nu; \theta_1, \ldots, \theta_L) = \frac{1}{L}\sum_{l=1}^{L} W_p^p(\theta_l \sharp \mu, \theta_l \sharp \nu). \tag{5}$$

We refer to Algorithm 2 in Appendix B for the computational algorithm of the QSW distance.

**Quasi-Sliced Wasserstein variants.** We refer to (i) QSW with **G**aussian-based mapping QMC point set as **GQSW**, (ii) QSW with **e**qual area mapping QMC point set as **EQSW**, (iii) QSW with QMC generalized **s**piral points as **SQSW**, (iv) QSW with maximizing **d**istance QMC point sets as **DQSW**, and (v) QSW with minimizing **C**oulomb energy sequence as **CQSW**.

**Proposition 1.** *With point sets constructed through the Gaussian-based mapping, the equal area mapping, the generalized spiral points, and minimizing Coulomb energy, we have $\widehat{QSW}_p^p(\mu, \nu; \theta_1, \ldots, \theta_L) \to SW_p^p(\mu, \nu)$ as $L \to \infty$.*

The proof of Proposition 1 is in Appendix A.1. We now discuss some properties of QSW variants.

**Computational complexities.** QSW variants are deterministic, which means that the construction of QMC point sets, which can be reused multiple times, carries a one-time cost. Therefore, the computation of QSW variants has the same properties as for the SW distance, i.e., the time and space complexities are $\mathcal{O}(Ln\log n + Ldn)$ and $\mathcal{O}(Ld + Ln)$, respectively. Since the QSW distance does not require resampling the set of projecting directions at each evaluation time, it is faster to compute than the SW distance if QMC point sets have been constructed in advance.

**Gradient Approximation.** When dealing with parametric probability measures, e.g., $\nu_\phi$, we might be interested in computing the gradient $\nabla_\phi SW_p^p(\mu, \nu_\phi)$ for optimization purposes. When using QMC integration, we obtain the corresponding deterministic approximation $\nabla_\phi \widehat{QSW}_p^p(\mu, \nu_\phi; \theta_1, \ldots, \theta_L) = \frac{1}{L}\sum_{l=1}^{L} \nabla_\phi W_p^p(\theta_l \sharp \mu, \theta_l \sharp \nu_\phi)$ for a QMC point set $\theta_1, \ldots, \theta_L$. For a more detailed definition of the gradient of the SW distance, please refer to Tanguy (2023). Since a deterministic gradient approximation may not lead to good convergence of optimization algorithms for relatively small $L$, we develop an unbiased estimation from QMC point sets in the next Section.

**Related works.** The SW distance is used as an optimization objective to construct a QMC point set on the unit cube and the unit ball in Paulin et al. (2020). However, a QMC point set on the unit-hypersphere is not discussed, and the SW distance is still approximated by conventional Monte Carlo integration. In contrast to the mentioned work, our focus is on using QMC point sets on the unit-hypersphere to approximate SW. The usage of heuristic scaled mapping with Halton sequence for SW distance approximation is briefly mentioned for the comparison between two Gaussians in (Lin et al., 2020). In this work, we consider a broader class of QMC point sets, assess their quality with the spherical cap discrepancy, discuss some randomized versions, and compare them in real applications. For further discussion on related work, please refer to Appendix C.

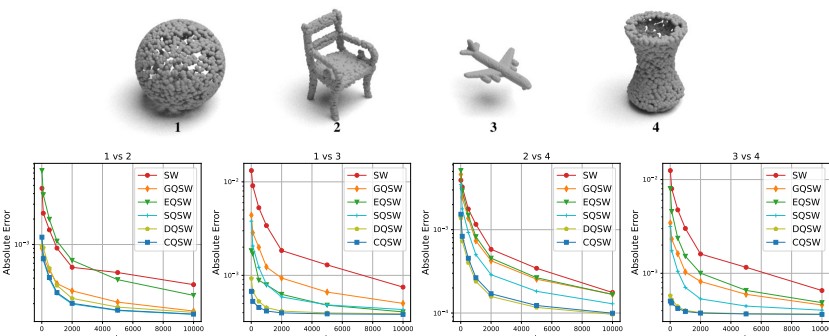

Figure 1: The error for approximation SW distances between empirical distributions over point-clouds.

### 3.3 RANDOMIZED QUASI-SLICED WASSERSTEIN

While QSW approximations could improve approximation error, they are all deterministic. Furthermore, the gradient estimator based on QSW is deterministic, which may not be well-suited for convergence in optimization with the SW loss function. Moreover, QSW cannot yield any confidence interval about the SW value. Consequently, we propose Randomized Quasi-Sliced Wasserstein estimations by introducing randomness into QMC point sets.

**Randomized Quasi-Monte Carlo methods.** The idea behind the Randomized Quasi-Monte Carlo (RQMC) approach is to inject randomness into a given QMC point set. For the unit cube, we can achieve a random QMC point set $x_1, \ldots, x_L$ by shifting (Cranley & Patterson, 1976) i.e., $y_1 = (x_i + U) \mod 1$ for all $i = 1, \ldots, L$ and $U \sim \mathcal{U}([0, 1]^d)$. In practice, scrambling (Owen, 1995) is preferable since it gives a uniformly distributed random vector when applied to $x \in [0, 1]^d$. In greater detail, $x$ is rewritten into $x = \sum_{k=1}^{\infty} b^{-k} a_k$ for base $b$ digits and $a_k \in \{0, 1, \ldots, b-1\}$. After that, we permute $a_1, \ldots, a_k$ randomly to obtain the scrambled version of $x$. Scrambling is applied to all points in a QMC point set to obtain a randomized QMC point set.

**Randomized QMC point sets on $\mathbb{S}^{d-1}$.** To the best of our knowledge, there is no prior work of randomized QMC point sets on the unit-hypersphere. Therefore, we discuss two practical ways to obtain random QMC point sets i.e., *pushfoward QMC point sets* and *random rotation*.

*Pushfoward QMC point sets.* Given a randomized QMC point set $x'_1, \ldots, x'_L$ on the unit-cube (unit-grid), we can use the Gaussian-based mapping (or the equal area mapping) to create a random QMC point set on the unit hypersphere $\theta'_1, \ldots, \theta'_L$. As long as the randomized sequence $x'_1, \ldots, x'_L$ is low-discrepancy on the mapping domain (e.g., as it happens when using scrambling), the spherical point set $\theta'_1, \ldots, \theta'_L$ will have the same uniformity as the non-randomized construction.

*Random rotation.* Given a QMC point set $\theta_1, \ldots, \theta_L$ on the unit-hypersphere $\mathbb{S}^{d-1}$, we can apply uniform random rotation to achieve a random QMC point set. In particular, we first sample $U \sim \mathcal{U}(\mathbb{V}_d(\mathbb{R}^d))$ where $\mathbb{V}_d(\mathbb{R}^d) = \{U \in \mathbb{R}^{d \times d} | U^\top U = I_d\}$ is the Stiefel manifold. After that, we form the new sequence $\theta'_1, \ldots, \theta'_L$ with $\theta'_i = U\theta_i$ for all $i = 1, \ldots, L$. Since rotation does not change the norm of vectors, the randomized QMC point set can be still a low-discrepancy sequence of the original QMC point set is low-discrepancy. Moreover, sampling uniformly from the Stiefel manifold is equivalent to applying the Gram-Smith orthogonalization process to $z_1, \ldots, z_l \overset{\text{iid}}{\sim} \mathcal{N}(0, I_d)$ by the Bartlett decomposition theorem (Muirhead, 2009).

**Definition 2.** *Given $p \geq 1$, $d \geq 2$, two measures $\mu, \nu \in \mathcal{P}_p(\mathbb{R}^d)$, and a randomized QMC point set $\theta'_1, \ldots, \theta'_L \in \mathbb{S}^{d-1}$, Randomized Quasi-Sliced Wasserstein estimation of order $p$ between $\mu$ and $\nu$ is:*

$$\widehat{RQSW}_p^p(\mu, \nu; \theta'_1, \ldots, \theta'_L) = \frac{1}{L} \sum_{l=1}^{L} W_p^p(\theta'_l \sharp \mu, \theta'_l \sharp \nu). \tag{6}$$

We refer to Algorithms 3 and 4 for more details on the computation of the RQSW approximation.

**Randomized Quasi-Sliced Wasserstein variants.** For pushfoward QMC point sets, we refer to (i) RQSW with **G**aussian-based mapping as **RGQSW**, (ii) RQSW with **e**qual area mapping as **REQSW**. For random rotation QMC point sets, we refer to (iii) RQSW with **G**aussian-based mapping as **RRGQSW**, (iv) RQSW with **e**qual area mapping as **RREQSW** (v) RQSW with generalized **s**piral

Table 1: Summary of Wasserstein-2 distances (multiplied by $10^2$) from three different runs.

| Estimators | Step 100 ($W_2 \downarrow$) | Step 200 ($W_2 \downarrow$) | Step 300 ($W_2 \downarrow$) | Step 400($W_2 \downarrow$) | Step 500 ($W_2 \downarrow$) | Time (s$\downarrow$) |
|---|---|---|---|---|---|---|
| SW | $5.761 \pm 0.088$ | $0.178 \pm 0.001$ | $0.025 \pm 0.001$ | $0.01 \pm 0.001$ | $0.004 \pm 0.001$ | 8.57 |
| GQSW | $6.136 \pm 0.0$ | $0.255 \pm 0.0$ | $0.077 \pm 0.0$ | $0.07 \pm 0.0$ | $0.068 \pm 0.0$ | 8.38 |
| EQSW | $\mathbf{5.414 \pm 0.0}$ | $0.22 \pm 0.0$ | $0.079 \pm 0.0$ | $0.071 \pm 0.0$ | $0.069 \pm 0.0$ | 8.37 |
| SQSW | $5.718 \pm 0.0$ | $0.181 \pm 0.0$ | $0.075 \pm 0.0$ | $0.07 \pm 0.0$ | $0.069 \pm 0.0$ | 8.38 |
| DQSW | $5.792 \pm 0.0$ | $0.193 \pm 0.0$ | $0.077 \pm 0.0$ | $0.07 \pm 0.0$ | $0.067 \pm 0.0$ | 8.37 |
| CQSW | $5.609 \pm 0.0$ | $\mathbf{0.163 \pm 0.0}$ | $0.07 \pm 0.0$ | $0.066 \pm 0.0$ | $0.065 \pm 0.0$ | 8.37 |
| RGQSW | $5.727 \pm 0.035$ | $0.169 \pm 0.003$ | $0.022 \pm 0.001$ | $0.007 \pm 0.001$ | $0.003 \pm 0.001$ | 8.75 |
| RRGQSW | $5.733 \pm 0.027$ | $0.168 \pm 0.006$ | $0.025 \pm 0.003$ | $0.011 \pm 0.002$ | $0.006 \pm 0.001$ | 8.49 |
| REQSW | $5.737 \pm 0.017$ | $0.171 \pm 0.004$ | $0.022 \pm 0.002$ | $0.007 \pm 0.001$ | $0.003 \pm 0.001$ | 8.78 |
| RREQSW | $5.704 \pm 0.011$ | $0.165 \pm 0.004$ | $0.021 \pm 0.0$ | $0.007 \pm 0.001$ | $0.003 \pm 0.001$ | 8.41 |
| RSQSW | $5.722 \pm 0.0$ | $0.169 \pm 0.001$ | $0.021 \pm 0.001$ | $0.007 \pm 0.001$ | $\mathbf{0.002 \pm 0.0}$ | 8.43 |
| RDQSW | $5.725 \pm 0.002$ | $0.169 \pm 0.001$ | $0.023 \pm 0.002$ | $0.009 \pm 0.002$ | $0.003 \pm 0.002$ | 8.44 |
| RCQSW | $5.721 \pm 0.002$ | $0.167 \pm 0.002$ | $\mathbf{0.02 \pm 0.0}$ | $\mathbf{0.007 \pm 0.001}$ | $0.003 \pm 0.001$ | 8.45 |

points as **RSQSW**, (vi) RQSW with maximizing **d**istance QMC point set as **RDQSW**, and (vii) RQSW with minimizing **C**oulomb energy sequence as **RCQSW**.

**Proposition 2.** *Gaussian-based mapping and random rotation randomized Quasi-Monte Carlo point sets are uniformly distributed, and the corresponding estimators $RQSW_p^p(\mu, \nu; \theta_1', \ldots, \theta_L')$ are unbiased estimations of $SW_p^p(\mu, \nu)$ i.e., $\mathbb{E}[\widehat{RQSW}_p^p(\mu, \nu; \theta_1', \ldots, \theta_L')] = SW_p^p(\mu, \nu)$.*

The proof of Proposition 2 is in Appendix A.2. We now discuss some properties of RQSW variants.

**Computational complexities.** Compared to QSW, RQSW requires additional computation for randomization. For the push-forward approach, scrambling and shifting carry a $\mathcal{O}(Ld)$ time complexity. In addition, mapping the randomized sequence from the unit-cube (unit-grid) to the unit-hypersphere has time complexity $\mathcal{O}(Ld)$. For the random rotation approach, sampling a random rotation matrix costs $\mathcal{O}(d^3)$. After that, multiplying the sampled rotation matrix with the precomputed QMC point set costs $\mathcal{O}(Ld^2)$ in time complexity and $\mathcal{O}(Ld)$ in space complexity. Overall, in the 3D setting where $d = 3$ and $n >> L > d$, the additional computation for RQSW approximations is negligible compared to the $\mathcal{O}(n \log n)$ cost from computing one-dimensional Wasserstein distances.

**Gradient estimation.** In contrast to QSW, RQSW is random and is an unbiased estimation when combined with the proposed construction of randomized QMC point sets from Proposition 2. Therefore, it follows directly that $\mathbb{E}[\nabla_\phi \widehat{RQSW}_p^p(\mu, \nu_\phi; \theta_1', \ldots, \theta_L')] = \nabla_\phi SW_p^p(\mu, \nu_\phi)$ due to the Leibniz rule of differentiation. Therefore, this estimation can lead to better convergence for optimization.

## 4 EXPERIMENTS

We first demonstrate that QSW variants outperform the conventional Monte Carlo approximation (referred to as SW) in Section 4.1. We then showcase the advantages of RQSW variants in point-cloud interpolation and image style transfer, comparing them to both QSW variants and the conventional SW approximation in Section 4.2 and Section 4.3, respectively. Finally, we present the favorable performance of QSW and RQSW variants in training a deep point-cloud autoencoder.

### 4.1 APPROXIMATION ERROR

**Setting.** We select randomly four point-clouds (1, 2, 3, and 4 with 3 dimensions, 2048 points) from ShapeNet Core-55 dataset (Chang et al., 2015) as shown in Figure 1. After that, we use MC estimation with $L = 100000$ to approximate $SW_2^2$ between empirical distributions over point-clouds 1-2, 1-3, 2-3, and 3-4, then treat them as the population value. Next, we vary $L$ in the set $\{10, 100, 500, 1000, 2000, 5000, 10000\}$ and compute the corresponding absolute error of the estimation from MC (SW), and QMC (QSWs).

**Results.** We illustrate the approximation errors in Figure 1. From the plot, it is evident that QSW approximations yield lower errors compared to the conventional SW approximation. Among the QSW approximations, CQSW and DQSW perform the best, followed by SQSW. In this simulation, the quality of GQSW and EQSW is not comparable to the previously mentioned approximations. Nevertheless, their errors are at least comparable to SW and are considerably better most of the time.

### 4.2 POINT-CLOUD INTERPOLATION

**Setting.** To interpolate between two point-clouds $X$ and $Y$, we define the curve $\dot{Z}(t) = -n \nabla_{Z(t)} \left[ SW_2 \left( P_{Z(t)}, P_Y \right) \right]$ where $P_X$ and $P_Y$ are empirical distributions over $X$ and $Y$ in turn. Here, the curve starts from $Z(0) = X$ and ends at $Y$. In this experiment, we set $X$ as point-cloud 1

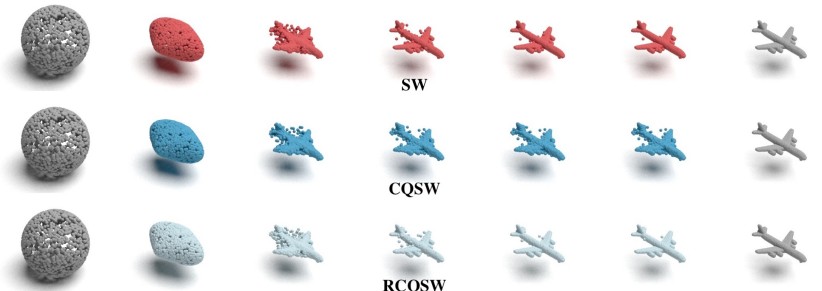

Figure 2: Point-cloud interpolation from SW, CQSW, and RCQSW with $L = 100$.

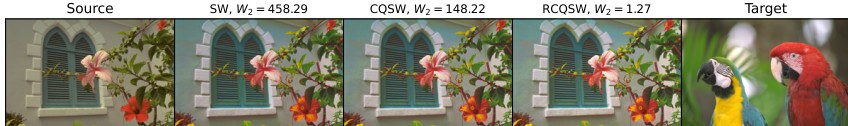

Figure 3: Style-transferred images from SW, CQSW, and RCQSW with $L = 100$.

Table 2: Reconstruction losses (multiplied by 100) from trained by different approximations with $L = 100$.

| Approximation | Epoch 100 | | Epoch 200 | | Epoch 400 | |
|---|---|---|---|---|---|---|
| | $SW_2(\downarrow)$ | $W_2(\downarrow)$ | $SW_2 (\downarrow)$ | $W_2(\downarrow)$ | $SW_2 (\downarrow)$ | $W_2(\downarrow)$ |
| SW | $2.25 \pm 0.06$ | $10.58 \pm 0.12$ | $2.11 \pm 0.04$ | $9.92 \pm 0.08$ | $1.94 \pm 0.06$ | $9.21 \pm 0.06$ |
| GQSW | $11.17 \pm 0.07$ | $32.58 \pm 0.06$ | $11.75 \pm 0.07$ | $33.27 \pm 0.09$ | $14.82 \pm 0.02$ | $37.99 \pm 0.05$ |
| EQSW | $2.25 \pm 0.02$ | $10.57 \pm 0.02$ | $2.05 \pm 0.02$ | $9.84 \pm 0.07$ | $1.90 \pm 0.04$ | $9.20 \pm 0.07$ |
| SQSW | $2.25 \pm 0.01$ | $10.57 \pm 0.03$ | $2.08 \pm 0.01$ | $9.90 \pm 0.04$ | $1.90 \pm 0.02$ | $9.17 \pm 0.05$ |
| DQSW | $2.24 \pm 0.07$ | $10.58 \pm 0.05$ | $2.06 \pm 0.04$ | $9.83 \pm 0.01$ | $1.86 \pm 0.05$ | $9.12 \pm 0.07$ |
| CQSW | $2.22 \pm 0.02$ | $10.54 \pm 0.02$ | $2.05 \pm 0.06$ | $\mathbf{9.81 \pm 0.04}$ | $\mathbf{1.84 \pm 0.02}$ | $\mathbf{9.06 \pm 0.02}$ |
| RGQSW | $2.25 \pm 0.02$ | $10.57 \pm 0.01$ | $2.09 \pm 0.03$ | $9.92 \pm 0.01$ | $1.94 \pm 0.02$ | $9.18 \pm 0.02$ |
| RRGQSW | $2.23 \pm 0.01$ | $10.51 \pm 0.04$ | $2.06 \pm 0.05$ | $9.84 \pm 0.06$ | $1.88 \pm 0.09$ | $9.16 \pm 0.11$ |
| REQSW | $2.24 \pm 0.04$ | $10.53 \pm 0.04$ | $2.08 \pm 0.04$ | $9.90 \pm 0.08$ | $1.89 \pm 0.04$ | $9.17 \pm 0.06$ |
| RREQSW | $2.21 \pm 0.04$ | $10.50 \pm 0.04$ | $2.03 \pm 0.02$ | $9.83 \pm 0.02$ | $1.88 \pm 0.05$ | $9.15 \pm 0.06$ |
| RSQSW | $2.22 \pm 0.05$ | $10.53 \pm 0.01$ | $2.04 \pm 0.06$ | $9.82 \pm 0.06$ | $1.85 \pm 0.05$ | $9.12 \pm 0.02$ |
| RDQSW | $\mathbf{2.21 \pm 0.03}$ | $\mathbf{10.50 \pm 0.02}$ | $2.03 \pm 0.04$ | $9.82 \pm 0.04$ | $1.86 \pm 0.03$ | $9.12 \pm 0.02$ |
| RCQSW | $2.22 \pm 0.03$ | $10.50 \pm 0.05$ | $\mathbf{2.03 \pm 0.02}$ | $9.82 \pm 0.03$ | $1.85 \pm 0.06$ | $9.12 \pm 0.03$ |

and Y as point-cloud 3 in Figure 1. After that, we use different gradient approximations from the conventional SW, QSW variants, and RQSW variants to perform the Euler scheme with 500 iterations, step size $0.01$. To verify which approximation gives the shortest curve in length, we compute the Wasserstein-2 distance (POT library, Flamary et al. (2021)) between $P_{Z(t)}$ and $P_Y$.

**Results.** We report Wasserstein-2 distances (from three different runs) between $P_{Z(t)}$ and $P_Y$ at time step $100, 200, 300, 400, 500$ in Table 1 with $L = 100$. From the table, we observe that QSW variants do not perform well in this application due to the deterministic approximation of the gradient with a fixed set of projecting directions. In particular, although EQSW and CQSW perform the best at time steps 100 and 200, QSW variants cannot make the curves terminate. As expected, RQSW variants can solve the issue by injecting randomness to create new *random* projecting directions. Compared to SW, RQSW variants are all better except RRGQSW. We visualize the interpolation for SW, CQSW, and RCQSW in Figure 2. The full visualization from all approximations is given in Figure 8 in Appendix D.2. From the figures, we observe that the qualitative comparison is consistent with the quantitative comparison in Table 1. In Appendix D.2, we also provide the result for $L = 10$ in Table 3, and the result for a different pair of point-clouds in Table 4-5 and Figure 9. We refer the reader to Appendix D.2 for a more detailed discussion.

### 4.3 IMAGE STYLE TRANSFER

**Setting.** Given a source image and a target image, we denote the associated color palettes as $X$ and $Y$, which are matrices of size $n \times 3$ ($n$ is the number of pixels). Similar to point-cloud interpolation, we iterate along the curve between $P_X$ and $P_Y$. However, since the value of the color palette (RGB) is in the set $\{0, \ldots, 255\}$, we need to perform an additional rounding step at the final Euler iterations. Moreover, we use more iterations i.e., 1000, and a bigger step size i.e., 1.

**Results.** For $L = 100$, we report the Wasserstein-2 distances at the final time step and the corresponding transferred images from SW, CQSW, and RCQSW in Figure 3. The full results for all approximations are given in Figure 10 in Appendix D.3. In addition, we provide results for $L = 10$ in Figure 11 in Appendix D.3. Overall, QSW variants and RQSW perform better than SW in terms

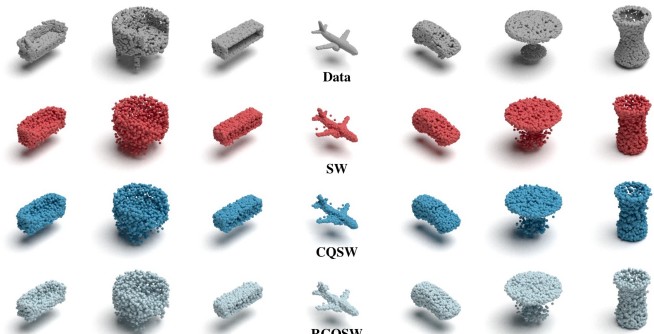

Figure 4: Reconstructed point-clouds from SW, CQSW, and RCQSW with $L = 100$.

of both Wasserstein distance and visualization (brighter transferred images). Comparing QSW and RQSW, the latter yields considerably lower Wasserstein distances. In this task, RQSW variants display quite similar performance. We refer the reader to Appendix D.3 for more detail.

## 4.4 DEEP POINT-CLOUD AUTOENCODER

**Setting.** We follow the experimental setting in (Nguyen et al., 2023) to train deep point-cloud autoencoders with the SW distance on the ShapeNet Core-55 dataset Chang et al. (2015). We aim to optimize the following objective $\min_{\phi,\gamma} \mathbb{E}_{X \sim \mu(X)}[\text{SW}_p(P_X, P_{g_\gamma(f_\phi(X))})]$, where $\mu(X)$ is our data distribution, $f_\phi$ and $g_\psi$ are a deep encoder and a deep decoder with Point-Net Qi et al. (2017) architecture. To optimize the objective, we use conventional MC estimation, QSW, and RQSW to approximate the gradient $\nabla_\phi$ and $\nabla_\psi$. We then utilize the standard SGD optimizer to train the autoencoder (with an embedding size of 256) for 400 epochs with a learning rate of 1e-3, a batch size of 128, a momentum of 0.9, and a weight decay of 5e-4. To evaluate the quality of trained autoencoders, we compute the average reconstruction losses, which are the $W_2$ and $SW_2$ distances (estimated with 10000 MC samples), on a different dataset i.e., ModelNet40 dataset (Wu et al., 2015).

**Results.** We report the reconstruction losses with $L = 100$ in Table 2 (from three different training times). Interestingly, CQSW performs the best among all approximations i.e., SW, QSW variants, and RQSW variants at the last epoch. We have an explanation for this phenomenon. In contrast to point-cloud interpolation which considers only one pair of point-clouds, we estimate an autoencoder from an entire dataset of point-clouds. Therefore, model misspecification might happen here i.e., the family of Point-Net autoencoders may not contain the true data-generating distribution. Hence, $L = 100$ might be large enough to approximate well with QSW. When we reduce $L$ to 10 in Table 6 in Appendix 4.4, CQSW and other QSW variants become considerably worse. In this application, we also observe that GQSW suffers from some numerical issues which leads to a very poor performance. As a solution, RQSW performs consistently well compared to SW especially random rotation variants. We present some reconstructed point-clouds from SW, CQSW, and RCQSW in Figure 4 and full visualization in Figure 12- 13. Overall, we recommend RCQSW for this task as a safe choice. We refer the reader to Appendix D.4 for more detail.

## 5 CONCLUSION

We presented Quasi-Sliced Wasserstein (QSW) approximation methods, which give rise to a better class of numerical estimates for the Sliced Wasserstein (SW) distance based on Quasi-Monte Carlo (QMC) methods. We discussed various ways to construct QMC point sets on the unit hypersphere, including the Gaussian-based mapping, the equal area mapping, generalized spiral points, maximizing distance points, and minimizing Coulomb energy points. Moreover, we proposed Randomized Quasi-Sliced Wasserstein (RQSW) approximations, which is a family of unbiased estimators of the SW distance based on injecting randomness into deterministic QMC point sets. We showed that QSW methods can reduce approximation error in comparing 3D point clouds. In addition, we showed that QSW variants and RQSW variants provide better gradient approximation for point-cloud interpolation, image-style transfer, and training point-cloud autoencoders. Overall, we recommend RQSW with random rotation of QMC point sets minimizing Coulomb energy, since it gives consistent and stable behavior across tested applications. In the future, we plan on extending QSW and RQSW approximations to higher dimensions $d > 3$, and apply QMC to other variants of the SW distance.

ACKNOWLEDGEMENTS

We would like to thank Peter Müller for his insightful discussion during the course of this project. NH acknowledges support from the NSF IFML 2019844 and the NSF AI Institute for Foundations of Machine Learning. NB acknowledges the financial support by the Bank of Italy's "G. Mortara" scholarship.

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

# Supplement to "Quasi-Monte Carlo for 3D Sliced Wasserstein"

We first provide proofs for theoretical results in the main text in Appendix A. Next, we offer additional background information, including the Wasserstein distance, and computational algorithms for SW, QSW, and RQSW variants in Appendix B. We then discuss related work in Appendix C. Afterward, we present detailed experimental results, which are mentioned in the main text for point-cloud interpolation, image style transfer, and deep point-cloud autoencoders in Appendix D. Finally, we report on the computational infrastructure in Appendix E

## A  PROOFS

### A.1  PROOF OF PROPOSITION 1

We first discuss the asymptotic uniformity of the mentioned QMC point set.

For the Gaussian-based mapping construction, From the construction, we have the function $\theta = f(x) = \frac{\Phi^{-1}(x)}{||\Phi^{-1}(x)||_2}$. Given a Sobol sequence $x_1, \ldots, x_L$, the corresponding spherical vectors are $\theta_1, \ldots, \theta_L$ with $\theta_l = f(x_l)$ for all $l = 1, \ldots, L$. Let $X_L \sim \frac{1}{L} \sum_{l=1}^{L} \delta_{x_l}$. From the low-discrepancy sequence property of Sobol sequences (Sobol, 1967), we have that $X_L$ converges to $X \sim \mathcal{U}(0, 1)$ in distribution as $L \to \infty$. Since our function $f(x)$ is continuous on $[0, 1]^d$, using the continuous mapping theorem, we have that $\theta_L = f(X_L)$ converges to $f(X) \sim \mathcal{U}(\mathbb{S}^{d-1})$ in distribution as $L \to \infty$. For the equal area mapping construction, we refer the reader to Aistleitner et al. (2012) for the proof of uniformity. For the generalized spiral points construction, we refer the reader to Hardin et al. (2016) for the proof of uniformity of this construction. Minimizing Coulomb energy is proven to create an asymptotic uniform sequence in Götz (2000).

Now denote $\gamma_L = \frac{1}{L} \sum_{i=1}^{L} \delta_{\theta_i}$ and $\theta \sim \mathcal{U}(\mathbb{S}^{d-1})$. Given an asymptotically uniform point set $\theta_1, \ldots, \theta_L$, we have $\gamma_L \overset{w}{\to} \mathcal{U}(\mathbb{S}^{d-1})$ as $L \to \infty$, where $\overset{w}{\to}$ denotes weak convergence of probability measures. That is, $\mathbb{E}_{\theta \sim \gamma_L}[g(\theta)] \to \mathbb{E}_{\theta \sim \mathcal{U}(\mathbb{S}^{d-1})}[g(\theta)]$ for all bounded continuous functions $g$. Thus, by the definition of the SW distance and its QSW approximation, one is left to show that $\theta \mapsto W_p^p(\theta \sharp \mu, \theta \sharp \nu)$ is bounded and continuous for any two measures $\mu, \nu$ with finite $p^{th}$ moment. We show these properties for $W_p(\theta \sharp \mu, \theta \sharp \nu)$ and then invoke continuity of the real function $x \mapsto x^{1/p}$.

As for boundedness, one can use the Cauchy-Schwartz inequality on $\mathbb{R}^d$ to get

$$W_p(\theta \sharp \mu, \theta \sharp \nu) = \left( \inf_{\pi \in \Pi(\nu, \mu)} \int_{\mathbb{R}^d} |\theta^\top x - \theta^\top y|^p \pi(dx, dy) \right)^{1/p}$$

$$\leq \left( \inf_{\pi \in \Pi(\nu, \mu)} \int_{\mathbb{R}^d} \|x - y\|^p \pi(dx, dy) \right)^{1/p}$$

$$= W_p(\mu, \nu) < \infty$$

for all $\theta \in \mathbb{S}^{d-1}$. As for continuity, let $(\theta_t)_{t \geq 1}$ be a sequence on $\mathbb{S}^{d-1}$ converging to $\theta \in \mathbb{S}^{d-1}$. Then

$$|W_p(\theta_t \sharp \mu, \theta_t \sharp \nu) - W_p(\theta \sharp \mu, \theta \sharp \nu)| \leq |W_p(\theta_t \sharp \mu, \theta_t \sharp \nu) - W_p(\theta \sharp \mu, \theta_t \sharp \nu)|$$
$$+ |W_p(\theta \sharp \mu, \theta_t \sharp \nu) - W_p(\theta \sharp \mu, \theta \sharp \nu)|$$
$$\leq W_p(\theta \sharp \mu, \theta_t \sharp \mu) + W_p(\theta \sharp \nu, \theta_t \sharp \nu),$$

where the last inequality is a straightforward consequence of the triangle inequality applied to the metric $W_p$. Let $\lambda \in \{\mu, \nu\}$. Then, using again the Cauchy-Schwartz inequality, we obtain

$$W_p(\theta \sharp \lambda, \theta_t \sharp \lambda) = \left( \inf_{\pi \in \Pi(\lambda, \lambda)} \int_{\mathbb{R}^d} |\theta_t^\top x - \theta^\top y|^p \pi(dx, dy) \right)^{1/p}$$

$$\leq \left( \int_{\mathbb{R}^d} |\theta_t^\top x - \theta^\top x|^p \lambda(dx) \right)^{1/p}$$

$$\leq \underbrace{\left( \int_{\mathbb{R}^d} \|x\|^p \lambda(dx) \right)^{1/p}}_{< \infty} \|\theta_t - \theta\| \to 0 \quad \text{as } t \to \infty.$$

This implies $\mathrm{W}_p(\theta_t\sharp\mu, \theta_t\sharp\nu) \to \mathrm{W}_p(\theta\sharp\mu, \theta\sharp\nu)$ as $t \to \infty$, proving continuity.

## A.2 PROOF OF PROPOSITION 2

For the Gaussian-based mapping construction, given a Sobol sequence $x_1, \ldots, x_L \in [0,1]^d$, applying scrambling returns $x'_1, \ldots, x'_L \in \mathcal{U}([0,1]^d)$ (Owen, 1995). Since $f(x) = \frac{\Phi^{-1}(x)}{||\Phi^{-1}(x)||_2}$ is the normalized inverse Gaussian CDF, $\theta'_l = f(x'_l) \sim \mathcal{U}(\mathbb{S}^{d-1})$ for all $l = 1, \ldots, L$.

For the random rotation construction, given a fixed vector $\theta \in \mathbb{S}^{d-1}$ and $U = (u_1, \ldots, u_d) \sim \mathcal{U}(\mathbb{V}_d(\mathbb{R}^d))$, we now prove that $U\theta \sim \mathcal{U}(\mathbb{S}^{d-1})$. For any $U_1 \in \mathbb{V}_d(\mathbb{R}^d)$, we have $U_1 U = U_2$ with $U_2 \sim \mathcal{U}(\mathbb{V}_d(\mathbb{R}^d))$. Therefore, we have that $U\theta$ has the same distribution as $U_1 U\theta$. Since there is only one distribution on $\mathbb{S}^{d-1}$ is invariant to rotation (Theorem 3.7 in (Mattila, 1999)) which is the uniform distribution, $U\theta \sim \mathcal{U}(\mathbb{S}^{d-1})$. Therefore, we obtain that $\theta'_1, \ldots, \theta'_L$, generated by uniform random rotation of a point set $\theta_1, \ldots, \theta_L$, are uniformly distributed.

Now, given $\theta'_1, \ldots, \theta'_L \sim \mathcal{U}(\mathbb{S}^{d-1})$, we have

$$\mathbb{E}[\widehat{\mathrm{RQSW}}_p^p(\mu, \nu; \theta'_1, \ldots, \theta'_L)] = \mathbb{E}\left[\frac{1}{L}\sum_{l=1}^L \mathrm{W}_p^p(\theta'_l\sharp\mu, \theta'_l\sharp\nu)\right]$$
$$= \frac{1}{L}\sum_{l=1}^L \mathbb{E}[\mathrm{W}_p^p(\theta'_l\sharp\mu, \theta'_l\sharp\nu)]$$
$$= \frac{1}{L}\sum_{l=1}^L \mathrm{SW}_p^p(\mu, \nu) = \mathrm{SW}_p^p(\mu, \nu),$$

which completes the proof.

## B ADDITIONAL BACKGROUND

**Wasserstein distance.** Given two probability measures $\mu \in \mathcal{P}_p(\mathbb{R}^d)$ and $\nu \in \mathcal{P}_p(\mathbb{R}^d)$, and $p \geq 1$, the Wasserstein distance (Villani, 2008; Peyré & Cuturi, 2019) between $\mu$ and $\nu$ is

$$\mathrm{W}_p(\mu, \nu) = \left(\inf_{\pi \in \Pi(\mu,\nu)} \int_{\mathbb{R}^d \times \mathbb{R}^d} ||x - y||_p^p d\pi(x,y)\right)^{1/p}, \tag{7}$$

where $\Pi(\mu, \nu)$ is the set of all couplings whose marginals are $\mu$ and $\nu$. Considering the discrete case, namely, $\mu = \sum_{i=1}^n \alpha_i \delta_{x_i}$ and $\nu = \sum_{j=1}^n \beta_j \delta_{y_j}$ with $\sum_{i=1}^n \alpha_i = \sum_{j=1}^n \beta_j$, one obtains:

$$\mathrm{W}_p^p(\mu, \nu) = \min_{\pi \in \Pi(\alpha,\beta)} \sum_{i=1}^n \sum_{j=1}^n \pi_{ij}||x_i - y_j||_p^p, \tag{8}$$

where $\Pi(\mu, \nu) = \{\pi \in \mathbb{R}_+^{n \times n} | \pi\mathbf{1} = \alpha, \pi^\top\mathbf{1} = \beta\}$. Using linear programming, the computational complexity and memory complexity of the Wasserstein distance are $\mathcal{O}(n^3 \log n)$ and $\mathcal{O}(n^2)$.

**Algorithms.** We first introduce the computational algorithm for Monte Carlo estimation of SW in Algorithm 1. Next, we provide the algorithm for QMC approximation of SW in Algorithm 2. Finally, we present the algorithms for Randomized QMC estimation of the SW distance with scrambling and random rotation in Algorithms 3 and 4, respectively.

**Generation of Sobol sequence.** From (Joe & Kuo, 2003), for generating a Sobol point set $x_1, \ldots, x_L \in [0,1]^d$, we need to follow the following procedure. For the $j$-th point, we need to choose a primitive polynomial of some degree $s_l$ in the field $\mathbb{Z}_2$ (set of integer of module 2), that is:

$$z^{s_j} + a_{1,j}z^{s_j-1} + \ldots + a_{s_j-1,j}z + 1,$$

where the coefficients $a_{1,j}, \ldots, a_{s_j-1,j}$ are either 0 or 1. We then use $a_{1,j}, \ldots, a_{s_j-1,j}$ to define a sequence $m_{1,j}, m_{2,j}, \ldots m_{s_j,j}$ such that:

$$m_{k,j} = 2a_{1,j}m_{k-1,j} \oplus 2^2 a_{2,j}m_{k-2,j} \oplus \ldots \oplus 2^{s_j-1}a_{s_j-1,j}m_{k-s_j+1,j} \oplus 2^{s_j}m_{k-s_j,1} \oplus m_{k-s_j,j},$$

---

**Algorithm 1** Monte Carlo estimation of the Sliced Wasserstein distance.

---

**Input:** Probability measures $\mu$ and $\nu$, $p > 1$, and the number of projections $L$.
Set $\widehat{\mathrm{SW}}_p^p(\mu, \nu; L) = 0$
**for** $l = 1$ to $L$ **do**
    Sample $\theta_l \sim \mathcal{U}(\mathbb{S}^{d-1})$
    Compute $\widehat{\mathrm{SW}}_p^p(\mu, \nu; L) = \frac{1}{L} \sum_{l=1}^{L} \int_0^1 |F_{\theta_l \sharp \mu}^{-1}(z) - F_{\theta_l \sharp \nu}^{-1}(z)|^p dz$
**end for**
**Return:** $\widehat{\mathrm{SW}}_p^p(\mu, \nu; L)$

---

**Algorithm 2** Quasi-Monte Carlo approximation of the sliced Wasserstein distance.

---

**Input:** Probability measures $\mu$ and $\nu$, $p > 1$, QMC point set $\theta_1, \ldots, \theta_L \in \mathbb{S}^{d-1}$.
Set $\widehat{\mathrm{QSW}}_p^p(\mu, \nu; \theta_1, \ldots, \theta_L) = 0$
**for** $l = 1$ to $L$ **do**
    Compute $\widehat{\mathrm{QSW}}_p^p(\mu, \nu; \theta_1, \ldots, \theta_L) = \widehat{\mathrm{QSW}}_p^p(\mu, \nu; \theta_1, \ldots, \theta_L) + \frac{1}{L} \int_0^1 |F_{\theta_l \sharp \mu}^{-1}(z) - F_{\theta_l \sharp \nu}^{-1}(z)|^p dz$
**end for**
**Return:** $\widehat{\mathrm{QSW}}_p^p(\mu, \nu; \theta_1, \ldots, \theta_L)$

---

**Algorithm 3** Randomized Quasi-Monte Carlo estimation of the Sliced Wasserstein distance with scrambling.

---

**Input:** Probability measures $\mu$ and $\nu$, $p > 1$, QMC point set $x_1, \ldots, x_L \in [0, 1]^d$.
Scramble $x_1, \ldots, x_L$ to obtain $x_1', \ldots, x_L'$
Compute $\theta_1', \ldots, \theta_L' = f(x_1'), \ldots, f(x_L')$ for $f$ the Gaussian-based mapping or the equal area mapping.
Set $\widehat{\mathrm{RQSW}}_p^p(\mu, \nu; \theta_1', \ldots, \theta_L') = 0$
**for** $l = 1$ to $L$ **do**
    Compute $\widehat{\mathrm{RQSW}}_p^p(\mu, \nu; \theta_1', \ldots, \theta_L') = \widehat{\mathrm{RQSW}}_p^p(\mu, \nu; \theta_1', \ldots, \theta_L') + \frac{1}{L} \int_0^1 |F_{\theta_l' \sharp \mu}^{-1}(z) - F_{\theta_l' \sharp \nu}^{-1}(z)|^p dz$
**end for**
**Return:** $\widehat{\mathrm{RQSW}}_p^p(\mu, \nu; \theta_1', \ldots, \theta_L')$

---

**Algorithm 4** The Randomized Quasi-Monte Carlo estimation of sliced Wasserstein distance with random rotation.

---

**Input:** Probability measures $\mu$ and $\nu$, $p \geq 1$, QMC point set $\theta_1, \ldots, \theta_L \in \mathbb{S}^{d-1}$.
Sample $U \sim \mathcal{U}(\mathbb{V}_d(\mathbb{R}^d))$
Compute $\theta_1', \ldots, \theta_L' = U\theta_1, \ldots, U\theta_L$
Set $\widehat{\mathrm{RQSW}}_p^p(\mu, \nu; \theta_1', \ldots, \theta_L') = 0$
**for** $l = 1$ to $L$ **do**
    Compute $\widehat{\mathrm{RQSW}}_p^p(\mu, \nu; \theta_1', \ldots, \theta_L') = \widehat{\mathrm{RQSW}}_p^p(\mu, \nu; \theta_1', \ldots, \theta_L') + \frac{1}{L} \int_0^1 |F_{\theta_l' \sharp \mu}^{-1}(z) - F_{\theta_l' \sharp \nu}^{-1}(z)|^p dz$
**end for**
**Return:** $\widehat{\mathrm{RQSW}}_p^p(\mu, \nu; \theta_1', \ldots, \theta_L')$

---

for $k > s_j + 1$ and $\oplus$ is the bit-by-bit exclusive-OR operator. The initial values of $m_{1,j}, m_{2,j}, \ldots m_{s_j,j}$ are chosen freely such that $m_{k,j}, 1 \leq k \leq s_j$ is odd and less than $2^k$. After that, direction numbers $v_{1,j}, v_{2,j}, \ldots v_{s_j,j}$ are defined as:

$$v_{k,j} = \frac{m_{k,j}}{2^k}.$$

Finally, we have:

$$x_{l,j} = b_1 v_{1,j} \oplus b_2 v_{2,j} \oplus \ldots, \oplus b_{s_j} v_{s_j,j},$$

where $b_i$ is the $i$-th bit from the right when $l$ is written in binary ,i.e., , $(\ldots b_2 b_1)^2$ is the binary representation of $l$. For greater detail, we refer the reader to (Joe & Kuo, 2003) for more detailed and practical algorithms.

**Confidence Intervals.** Using the discussed methodology, one can obtain $M$ i.i.d RQSW estimates, i.e., $\widehat{\text{RQSW}}_p^p(\mu, \nu; \theta'_{1m}, \ldots, \theta'_{Lm})$ for $m = 1, \ldots, M$. Since RQSW is an unbiased estimate of the population SW, the central limit theorem ensures the following:

$$\frac{\hat{\mu}_M - \text{SW}_p^p(\mu, \nu)}{\hat{s}_M / \sqrt{M}} \xrightarrow{d} \mathcal{N}(0, 1)$$

as $M \to \infty$, where $\hat{\mu}_M$ and $\hat{s}_M$ are the sample mean and standard deviation based on the generated $M$-size sample. Therefore, a $1 - \alpha$ size asymptotic confidence interval for $\text{SW}_p^p(\mu, \nu)$ is readily obtained as

$$\hat{\mu}_M \pm z_{\alpha/2} \hat{s}_M / \sqrt{M}.$$

with $z_{\alpha/2}$ denoting the $\alpha/2$ quantile of a standard normal random variable. Alternatively, by sampling with replacement from the $M$ generated RQSW estimates, one can obtain $B$ bootstrap replications of $\hat{\mu}_M$, say $\hat{\mu}_M^{(b)}$ for $b = 1, \ldots, B$, and construct a $1 - \alpha$ bootstrap confidence interval for $\text{SW}_p^p(\mu, \nu)$ as $[\hat{q}_{\alpha/2}, \hat{q}_{1-\alpha/2}]$, where $\hat{q}_\omega$ denotes the $\omega$ sample quantile of $\{\hat{\mu}_M^{(b)} : b = 1, \ldots, B\}$.

## C    RELATED WORKS

**Beyond the uniform slicing distribution.** Recent works have explored non-uniform slicing distributions (Nguyen et al., 2021; Nguyen & Ho, 2023). Nevertheless, the uniform distribution remains foundational in constructing the pushforward slicing distribution (Nguyen et al., 2021) and the proposal distribution (Nguyen & Ho, 2023). Consequently, Quasi-Monte Carlo methods can also enhance the approximation of the uniform distribution.

**Beyond 3D.** It is worth noting that the Gaussian-based construction, maximizing distance, and minimizing Coulomb energy can be applied directly in higher dimensions, i.e., $d > 3$. Similarly, their randomized versions could also be used directly in higher dimensions. However, the quality of QMC point sets in high dimensions and their approximation errors are still open questions and require a detailed investigation. Therefore, we will leave this exploration to future work

**Scaled Mapping.** Quasi-Monte Carlo is briefly used for SW in (Lin et al., 2020). In particular, the authors utilize the Halton sequence in the three-dimensional unit cube, then map them to the unit sphere via the scaled mapping $f(x) = \frac{x}{\|x\|_2}$. However, this construction is heuristic and lacks meaningful properties. We visualize point sets of sizes $10, 50, 100$ in Figure 5. From the figure, it is evident that this construction does not exhibit low-discrepancy behavior, as all points are concentrated in one region of the sphere.

**Near Orthogonal Monte Carlo.** Motivated by orthogonal Monte Carlo, the authors in (Lin et al., 2020) propose near-orthogonal Monte Carlo, aiming to make the angles between any two samples close to orthogonal. We utilized the published code for optimization-based approaches available at `https://github.com/HL-hanlin/OMC` to generate point sets of size $L$ in three dimensions, where $L$ is chosen from the set $10, 50, 100$. For our experiments, we generated only one batch of $L$ points, avoiding the need to specify the second hyperparameter related to the number of batches. We visualize the resulting point sets and their corresponding spherical cap discrepancies in Figure 5. From the figure, it is evident that NOMC yields better spherical cap discrepancies compared to conventional Monte Carlo methods. However, it is important to note that NOMC does not achieve the same level of performance as the QMC point sets we discuss in this work. In this study, our primary focus is on QMC methods, and as such, we leave a detailed investigation of the application of OMC methods for SW to future research.

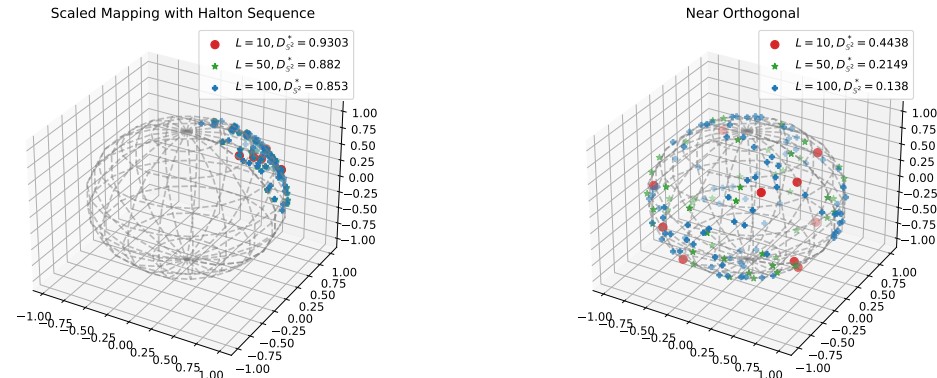

Figure 5: Optimization-based Orthogonal Monte Carlo point set and scaled mapping with Halton Sequence point set.

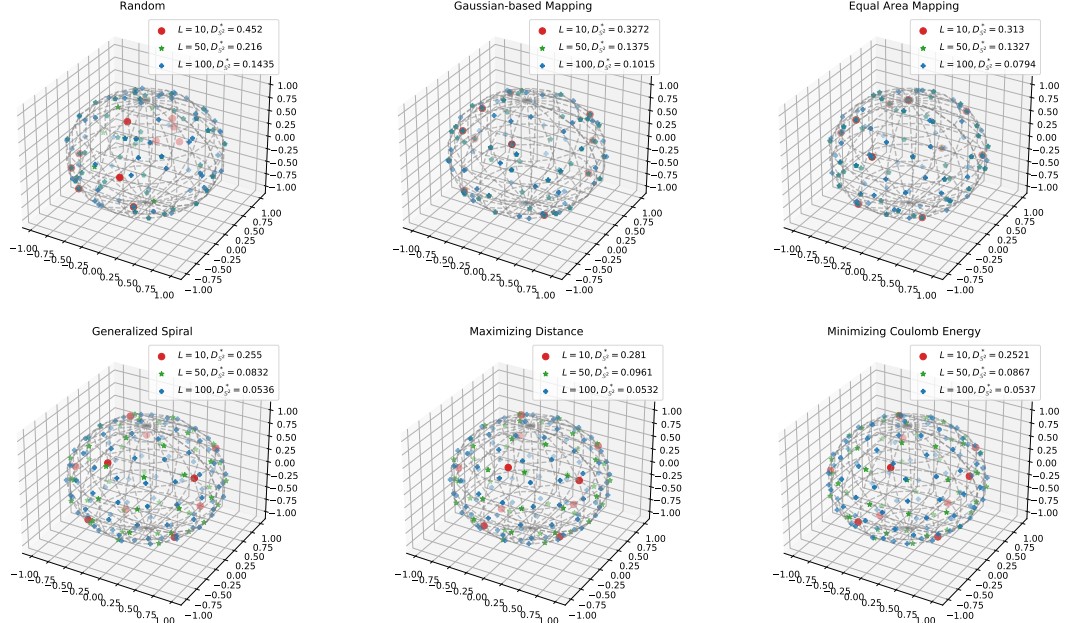

Figure 6: point sets on $\mathbb{S}^2$ with the size of $10, 50, 100$ and the corresponding spherical cap discrepancies.

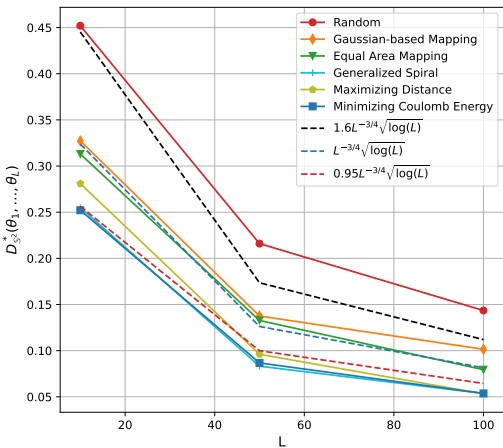

Figure 7: Spherical cap discrepancies of different QMC point sets and random point set.

Table 3: Summary of Wasserstein-2 distances (multiplied by $10^2$) from three different runs.

| Estimators | Step 100 ($W_2 \downarrow$) | Step 200 ($W_2 \downarrow$) | Step 300 ($W_2 \downarrow$) | Step 400 ($W_2 \downarrow$) | Step 500 ($W_2 \downarrow$) | Time (s$\downarrow$) |
|---|---|---|---|---|---|---|
| SW L=10 | $5.821 \pm 0.149$ | $0.203 \pm 0.012$ | $0.038 \pm 0.002$ | $0.017 \pm 0.001$ | $0.009 \pm 0.0$ | 2.90 |
| GQSW L=10 | $9.274 \pm 0.0$ | $3.776 \pm 0.0$ | $2.572 \pm 0.0$ | $2.297 \pm 0.0$ | $2.23 \pm 0.0$ | 2.69 |
| EQSW L=10 | $\mathbf{4.066 \pm 0.0}$ | $0.575 \pm 0.0$ | $0.511 \pm 0.0$ | $0.508 \pm 0.0$ | $0.508 \pm 0.0$ | 2.70 |
| SQSW L=10 | $6.321 \pm 0.0$ | $1.093 \pm 0.0$ | $0.603 \pm 0.0$ | $0.559 \pm 0.0$ | $0.554 \pm 0.0$ | 2.68 |
| DQSW L=10 | $5.919 \pm 0.0$ | $0.87 \pm 0.0$ | $0.607 \pm 0.0$ | $0.593 \pm 0.0$ | $0.593 \pm 0.0$ | 2.69 |
| CQSW L=10 | $5.561 \pm 0.0$ | $0.793 \pm 0.0$ | $0.614 \pm 0.0$ | $0.606 \pm 0.0$ | $0.606 \pm 0.0$ | 2.71 |
| RGQSW L=10 | $5.863 \pm 0.029$ | $\mathbf{0.188 \pm 0.007}$ | $0.035 \pm 0.002$ | $0.018 \pm 0.001$ | $0.01 \pm 0.001$ | 3.23 |
| RRGQSW L=10 | $5.781 \pm 0.102$ | $0.232 \pm 0.031$ | $0.047 \pm 0.002$ | $0.03 \pm 0.002$ | $0.026 \pm 0.001$ | 3.02 |
| REQSW L=10 | $5.733 \pm 0.19$ | $0.19 \pm 0.014$ | $\mathbf{0.034 \pm 0.003}$ | $0.016 \pm 0.002$ | $0.008 \pm 0.002$ | 3.12 |
| RREQSW L=10 | $5.857 \pm 0.058$ | $0.219 \pm 0.007$ | $0.042 \pm 0.001$ | $0.022 \pm 0.001$ | $0.014 \pm 0.001$ | 3.01 |
| RSQSW L=10 | $5.754 \pm 0.028$ | $0.195 \pm 0.004$ | $0.035 \pm 0.002$ | $0.016 \pm 0.002$ | $\mathbf{0.007 \pm 0.001}$ | 3.00 |
| RDQSW L=10 | $5.835 \pm 0.071$ | $0.202 \pm 0.011$ | $0.036 \pm 0.002$ | $\mathbf{0.016 \pm 0.001}$ | $0.008 \pm 0.001$ | 3.01 |
| RCQSW L=10 | $5.794 \pm 0.076$ | $0.196 \pm 0.008$ | $0.037 \pm 0.003$ | $0.017 \pm 0.002$ | $0.008 \pm 0.001$ | 3.02 |

# D DETAILED EXPERIMENTS

## D.1 SPHERICAL CAP DISCREPANCY

We plotted the spherical cap discrepancies of the discussed QMC point sets and added hypothetical lines of $CL^{-3/4}\sqrt{\log(L)}$ for $C = 1.6, C = 1, C = 0.95$ in Figure 7. From the figure, it is evident that QMC point sets derived from generalized spiral points, maximizing distance, and minimizing Coulomb energy exhibit a faster convergence rate than $\mathcal{O}(L^{-3/4}\sqrt{\log(L)})$. Consequently, they can be classified as low-discrepancy sequences. Regarding the equal-area mapping construction, it demonstrates approximately the same convergence rate as $\mathcal{O}(L^{-3/4}\sqrt{\log(L)})$, suggesting its potential as a low-discrepancy sequence. However, Gaussian-based mapping QMC point sets and random (MC) point sets do not exhibit low-discrepancy behavior. In summary, we recommend using generalized spiral points, maximizing distance, and minimizing Coulomb energy point sets for approximating SW when distance values are a critical factor in the application.

## D.2 POINT-CLOUD INTERPOLATION

**Approximate Euler methods.** We want to iterate through the curve $\dot{Z}(t) = -n\nabla_{Z(t)}\left[\text{SW2}\left(PZ(t), P_Y\right)\right]$. For each iteration with $t = 1, \ldots, T$, we first construct a point set $\theta_1, \ldots, \theta_L$ based on the discussed approaches using MC, QMC methods, and randomized QMC methods. After that, with a step size $\eta > 0$, we update:

$$Z(t) = Z(t-1) - n\eta\nabla_{Z(t-1)}\left[\frac{1}{L}\sum_{l=1}^{L}\text{W}_2^2\left(\theta_l\sharp P_{Z(t-1)}, \theta_l\sharp P_Y\right)\right]^{1/2}.$$

**Visualization for $L = 100$.** In addition to the partial visualization in the main text, we provide a full visualization of point-cloud interpolation from all QSW and RQSW variants in Figure 8. We observe that QSW variants cannot produce smooth point clouds at the final time step since they use the same QMC point sets across all time steps. In contrast, RQSW variants expedite the process of achieving a smooth point cloud that closely resembles the target. When compared to RQSW variants, the point cloud at the final time step from SW (the conventional MC) still contains some points that deviate significantly from the main shape.

**Results for $L = 10$.** We repeated the same experiments with $L = 10$. We have reported the Wasserstein-2 distances for intermediate point-clouds (relative to the target point-cloud) in Table 3. We observed a similar phenomenon as with $L = 100$, namely, RQSW outperforms QSW significantly and also performs better than SW. Compared to $L = 100$, all approximations from $L = 10$ yield higher Wasserstein-2 distances. However, the gaps between QSW variants are wider. Therefore, RQSW variants are more robust to the choice of $L$ than QSW.

**Results for a different pair of point-clouds.** We conduct the same experiments with a different pair of point-clouds, namely, 2 and 3 in Figure 1. We present a summary of Wasserstein-2 distances in Table 4 for $L = 100$ and Table 5 for $L = 10$. We observe the same phenomena as in the previous experiments. Firstly, RQSW variants produce shorter curves than QSW variants. Secondly, a larger

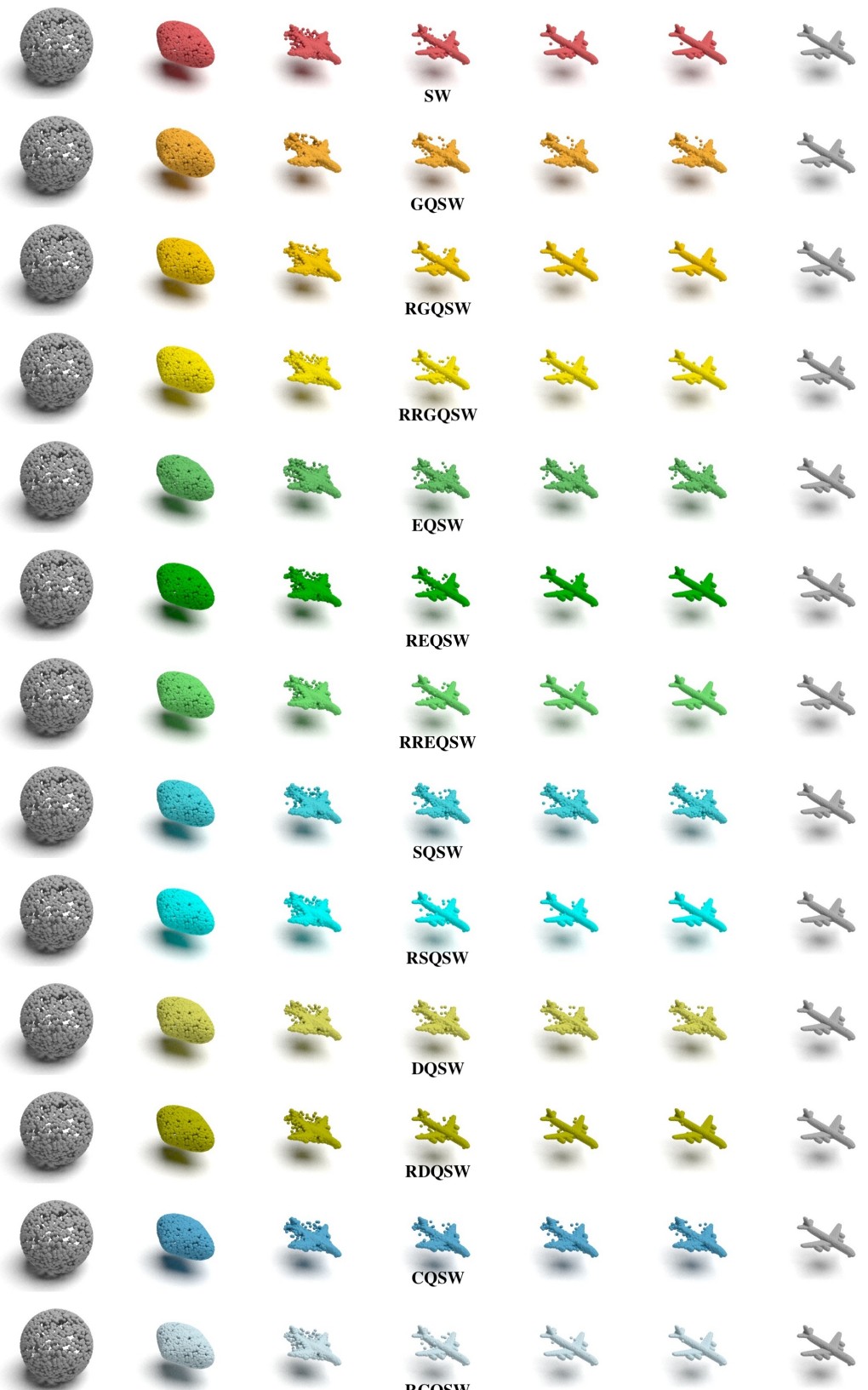

Figure 8: Point-cloud interpolation from SW, QSW variants, and RQSW variants with $L = 100$.

Table 4: Summary of Wasserstein-2 distances (multiplied by $10^2$) from three different runs.

| Estimators | Step 100 ($W_2 \downarrow$) | Step 200 ($W_2 \downarrow$) | Step 300 ($W_2 \downarrow$) | Step 400($W_2 \downarrow$) | Step 500 ($W_2 \downarrow$) |
|---|---|---|---|---|---|
| SW L=100 | $2.819 \pm 0.044$ | $0.23 \pm 0.002$ | $0.033 \pm 0.002$ | $0.012 \pm 0.002$ | $0.006 \pm 0.001$ |
| GQSW L=100 | $2.868 \pm 0.0$ | $0.281 \pm 0.0$ | $0.107 \pm 0.0$ | $0.093 \pm 0.0$ | $0.091 \pm 0.0$ |
| EQSW L=100 | $\mathbf{2.473 \pm 0.0}$ | $0.229 \pm 0.0$ | $0.109 \pm 0.0$ | $0.1 \pm 0.0$ | $0.098 \pm 0.0$ |
| SQSW L=100 | $2.841 \pm 0.0$ | $0.262 \pm 0.0$ | $0.109 \pm 0.0$ | $0.098 \pm 0.0$ | $0.096 \pm 0.0$ |
| DQSW L=100 | $2.883 \pm 0.0$ | $0.262 \pm 0.0$ | $0.101 \pm 0.0$ | $0.093 \pm 0.0$ | $0.091 \pm 0.0$ |
| CQSW L=100 | $2.696 \pm 0.0$ | $0.223 \pm 0.0$ | $0.092 \pm 0.0$ | $0.085 \pm 0.0$ | $0.084 \pm 0.0$ |
| RGQSW L=100 | $2.815 \pm 0.017$ | $0.231 \pm 0.005$ | $0.031 \pm 0.001$ | $0.01 \pm 0.001$ | $0.004 \pm 0.001$ |
| RRGQSW L=100 | $2.82 \pm 0.044$ | $0.233 \pm 0.008$ | $0.034 \pm 0.002$ | $0.013 \pm 0.002$ | $0.006 \pm 0.002$ |
| REQSW L=100 | $2.826 \pm 0.006$ | $0.229 \pm 0.002$ | $0.03 \pm 0.001$ | $0.01 \pm 0.0$ | $0.004 \pm 0.0$ |
| RREQSW L=100 | $2.83 \pm 0.015$ | $0.23 \pm 0.002$ | $0.031 \pm 0.001$ | $0.011 \pm 0.0$ | $0.005 \pm 0.001$ |
| RSQSW L=100 | $2.796 \pm 0.003$ | $\mathbf{0.224 \pm 0.001}$ | $0.028 \pm 0.002$ | $0.008 \pm 0.001$ | $0.003 \pm 0.0$ |
| RDQSW L=100 | $2.793 \pm 0.002$ | $\mathbf{0.224 \pm 0.001}$ | $\mathbf{0.028 \pm 0.001}$ | $\mathbf{0.008 \pm 0.001}$ | $\mathbf{0.002 \pm 0.0}$ |
| RCQSW L=100 | $2.794 \pm 0.005$ | $0.227 \pm 0.002$ | $0.03 \pm 0.001$ | $0.01 \pm 0.002$ | $0.005 \pm 0.002$ |

Table 5: Summary of Wasserstein-2 distances (multiplied by $10^2$) from three different runs.

| Estimators | Step 100 ($W_2 \downarrow$) | Step 200 ($W_2 \downarrow$) | Step 300 ($W_2 \downarrow$) | Step 400($W_2 \downarrow$) | Step 500 ($W_2 \downarrow$) |
|---|---|---|---|---|---|
| SW L=10 | $2.919 \pm 0.082$ | $0.262 \pm 0.018$ | $0.048 \pm 0.007$ | $0.02 \pm 0.004$ | $0.01 \pm 0.003$ |
| GQSW L=10 | $6.576 \pm 0.0$ | $2.863 \pm 0.0$ | $2.305 \pm 0.0$ | $2.197 \pm 0.0$ | $2.165 \pm 0.0$ |
| EQSW L=10 | $2.391 \pm 0.0$ | $0.789 \pm 0.0$ | $0.617 \pm 0.0$ | $0.6 \pm 0.0$ | $0.6 \pm 0.0$ |
| SQSW L=10 | $3.498 \pm 0.0$ | $1.437 \pm 0.0$ | $0.87 \pm 0.0$ | $0.783 \pm 0.0$ | $0.776 \pm 0.0$ |
| DQSW L=10 | $2.9 \pm 0.0$ | $1.118 \pm 0.0$ | $0.796 \pm 0.0$ | $0.754 \pm 0.0$ | $0.746 \pm 0.0$ |
| CQSW L=10 | $3.465 \pm 0.0$ | $1.596 \pm 0.0$ | $1.129 \pm 0.0$ | $1.035 \pm 0.0$ | $1.027 \pm 0.0$ |
| RGQSW L=10 | $2.979 \pm 0.048$ | $0.266 \pm 0.007$ | $0.045 \pm 0.002$ | $0.019 \pm 0.001$ | $0.009 \pm 0.001$ |
| RRGQSW L=10 | $2.928 \pm 0.056$ | $0.271 \pm 0.021$ | $0.051 \pm 0.003$ | $0.028 \pm 0.002$ | $0.022 \pm 0.001$ |
| REQSW L=10 | $2.891 \pm 0.089$ | $0.25 \pm 0.013$ | $0.045 \pm 0.002$ | $0.02 \pm 0.001$ | $0.01 \pm 0.001$ |
| RREQSW L=10 | $2.907 \pm 0.103$ | $0.268 \pm 0.011$ | $0.055 \pm 0.003$ | $0.027 \pm 0.002$ | $0.017 \pm 0.001$ |
| RSQSW L=10 | $\mathbf{2.747 \pm 0.006}$ | $0.24 \pm 0.002$ | $0.047 \pm 0.0$ | $0.02 \pm 0.001$ | $0.01 \pm 0.002$ |
| RDQSW L=10 | $2.769 \pm 0.015$ | $\mathbf{0.239 \pm 0.008}$ | $0.044 \pm 0.004$ | $0.019 \pm 0.004$ | $0.009 \pm 0.002$ |
| RCQSW L=10 | $2.761 \pm 0.101$ | $0.241 \pm 0.009$ | $\mathbf{0.043 \pm 0.0}$ | $\mathbf{0.018 \pm 0.002}$ | $\mathbf{0.009 \pm 0.001}$ |

number of projections is better, and RQSW variants are more robust to changes in $L$ than QSW variants. Additionally, we provide visualizations for $L = 100$ in Figure 9. From the figure, we can see consistent qualitative comparisons with the Wasserstein-2 distances reported in the tables.

**Recommended variants.** Overall, we recommend RSQSW, RDQSW, and RCQSW for the point-cloud interpolation application since they give consistent performance for $L = 100$ and $L = 10$ for both tried pairs of point-clouds.

### D.3 IMAGE STYLE TRANSFER

**Detailed settings.** We first reduce the number of colors in the images to 3000 using K-means clustering. Similar to the point-cloud interpolation, we iterate through the curve between the empirical distribution of colors in the source image and the empirical distribution of colors in the target image using the approximate Euler method.

**Full results for $L = 100$.** We present style-transferred images and their corresponding Wasserstein-2 distances to the target image in terms of color palettes at the last iteration (1000) in Figure 10. From the figure, it is evident that QSW variants facilitate faster color transfer compared to SW. To elaborate, SW exhibits a Wasserstein-2 distance of 458.29, while the highest Wasserstein-2 distance among QSW variants is 158.6, achieved by GQSW. The use of RQSW can further enhance quality; for instance, the highest Wasserstein-2 distance among RQSW variants is 1.45, achieved by RGQSW. The best-performing variant in this application is RSQSW; however, other RQSW variants are also comparable.

**Full results for $L = 10$.** We repeat the experiment with $L = 10$. In all approximations, decreasing $L$ to 10 results in a higher Wasserstein-2 distance, which is understandable based on the approximation error analysis. In this scenario, the performance of some QSW variants (GQSW, EBQSW, SQSW) degrades to the point of being even worse than SW. In contrast, the degradation of RQSW variants is negligible, particularly for RCQSW.

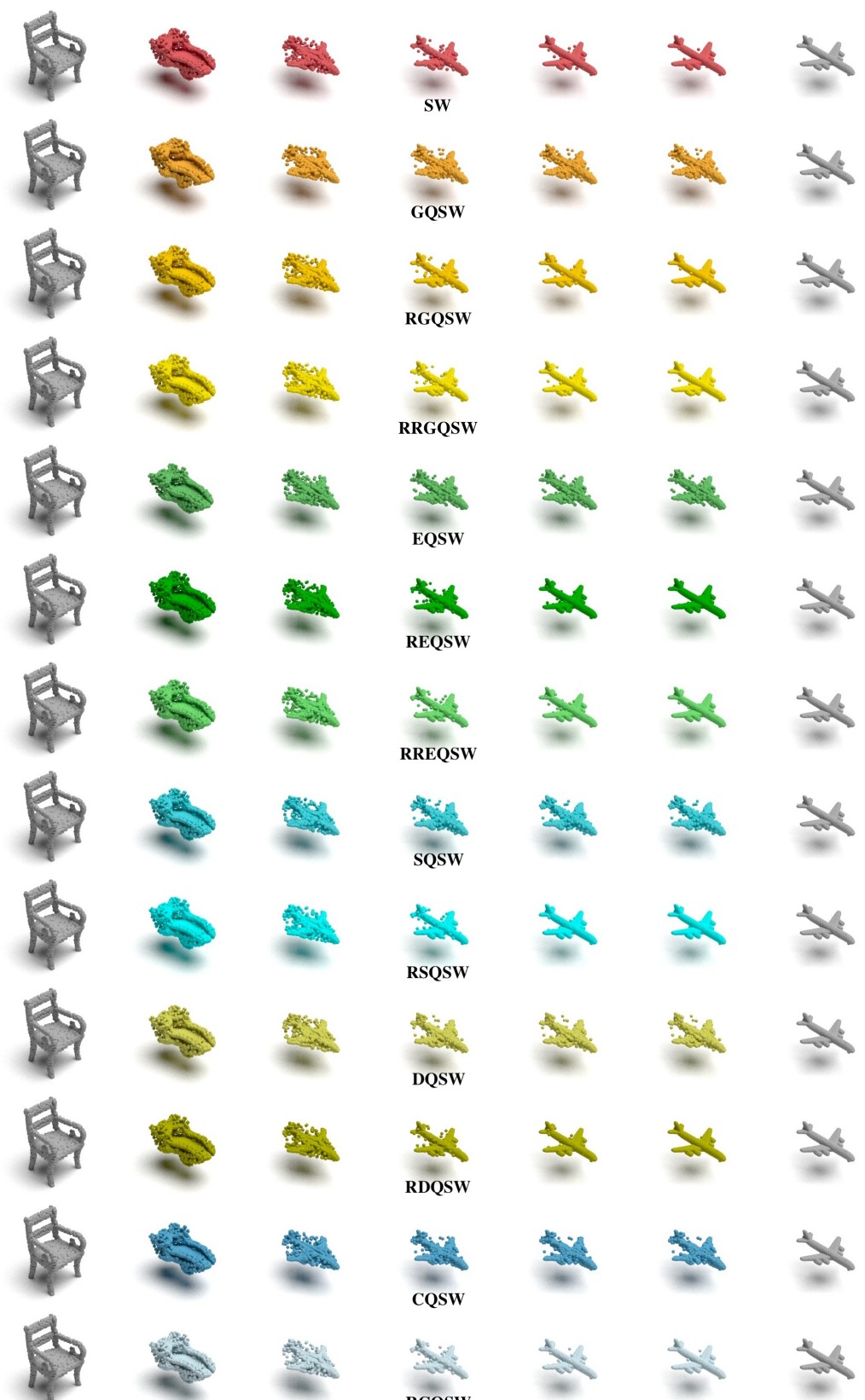

Figure 9: Point-cloud interpolation from SW, QSW variants, and RQSW variants with $L = 100$.

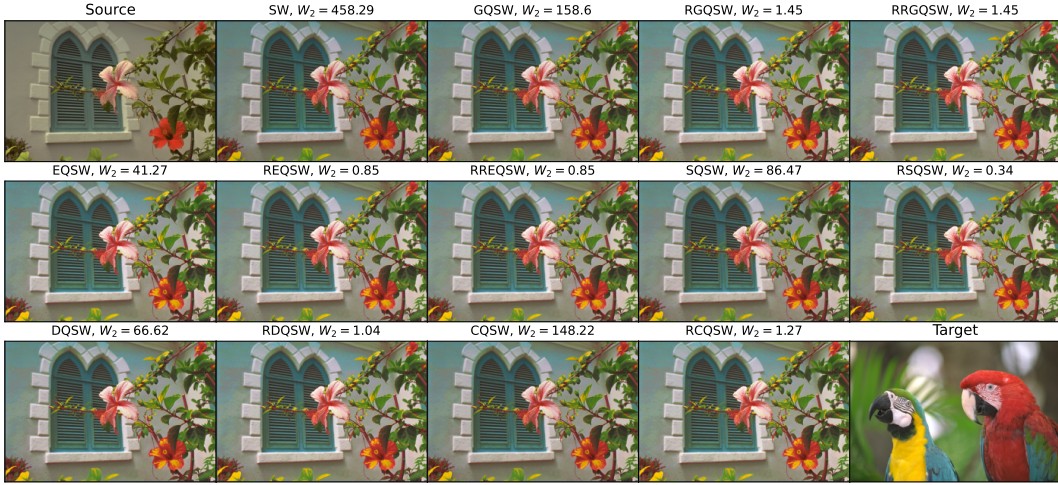

Figure 10: Style-transferred images from SW, QSW variants, and RQSW variants with $L = 100$.

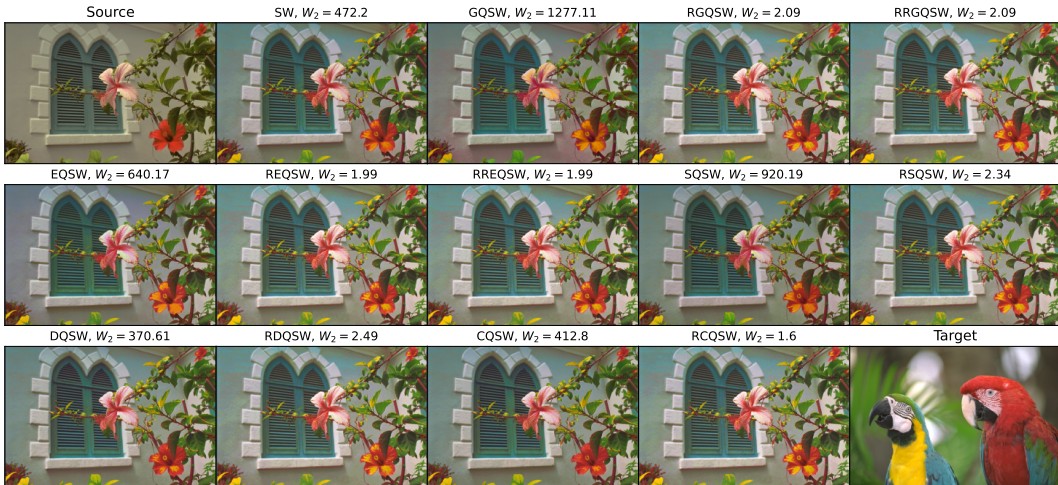

Figure 11: Style-transferred images from SW, QSW variants, and RQSW variants with $L = 100$.

**Recommended variants.** Overall, we recommend RCQSW for this application since it performs consistently for both setting of $L = 100$ and $L = 10$.

### D.4 DEEP POINT-CLOUD AUTOENCODER

**Full visualization for $L = 100$.** We first visualize reconstructed point-clouds from all approximations, including SW, QSW variants, and RQSW variants in Figure 12. Overall, we observe that the sharpness of the reconstructed point-clouds aligns with the reconstruction losses presented in Table 2. However, the point-clouds generated by GQSW lack meaningful structure, likely due to numerical issues encountered during training. These issues may stem from the numerical computation of the inverse CDF for specific projecting directions at certain iterations. Randomized versions of GQSW could potentially mitigate such problems, as stochastic gradient training may help avoid undesirable configurations in neural networks.

**Results for $L = 10$.** We reduce the number of projections $L$ to 10 and subsequently report the reconstruction losses in Table 6. Similar to other applications, reducing $L$ results in increased reconstruction losses, particularly for QSW variants. In this specific application, RQSW variants demonstrate their robustness to the choice of $L$; the reconstruction losses for $L = 10$ are comparable to those for $L = 100$, as shown in Table 2. Additionally, we provide visualizations of the reconstructed point-clouds for $L = 10$ in Figure 13. It is evident from the figure that reconstructed point-clouds from QSW variants exhibit significant noise.

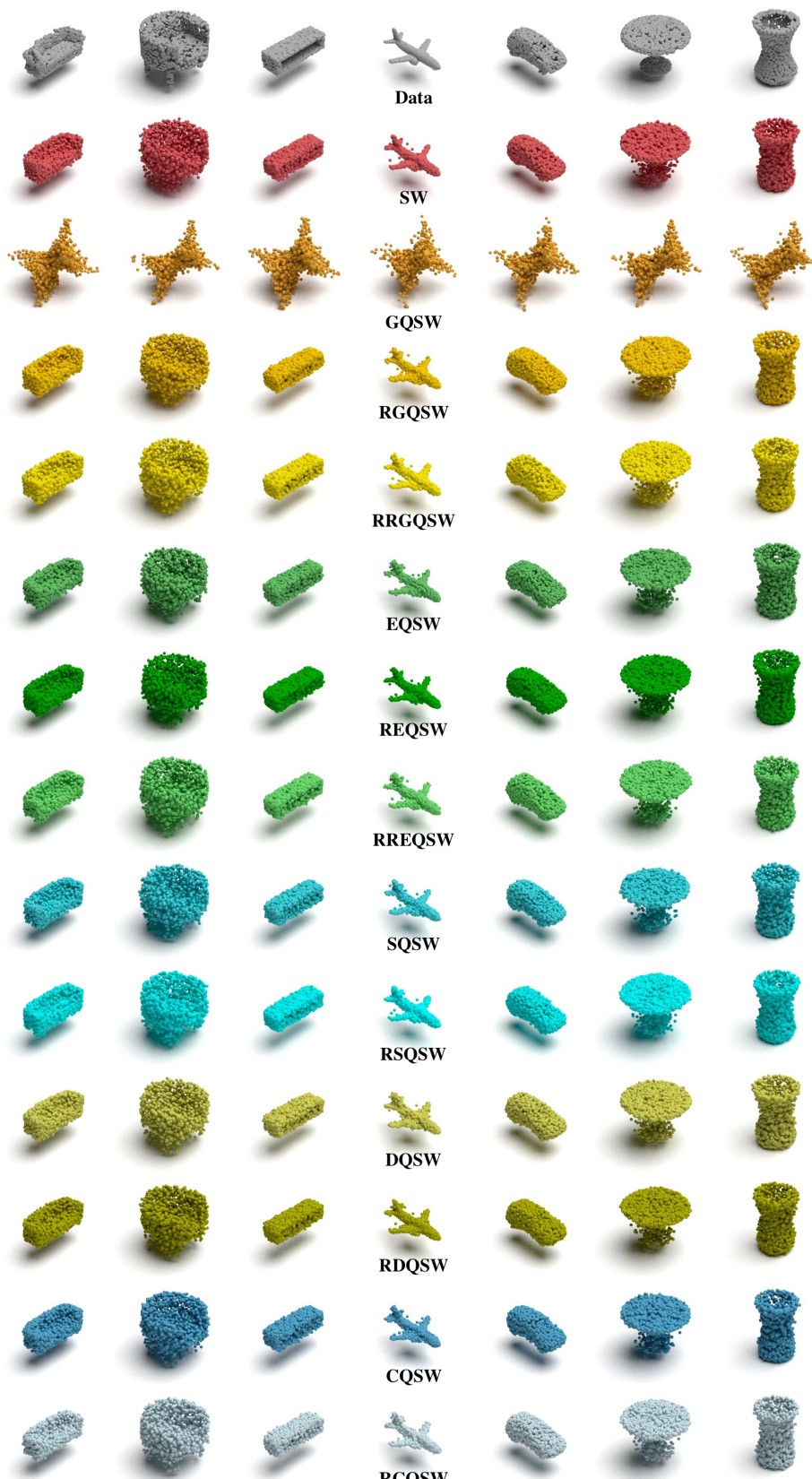

Figure 12: Some reconstructed point-clouds from SW, QSW variants, and RQSW variants with $L = 100$.

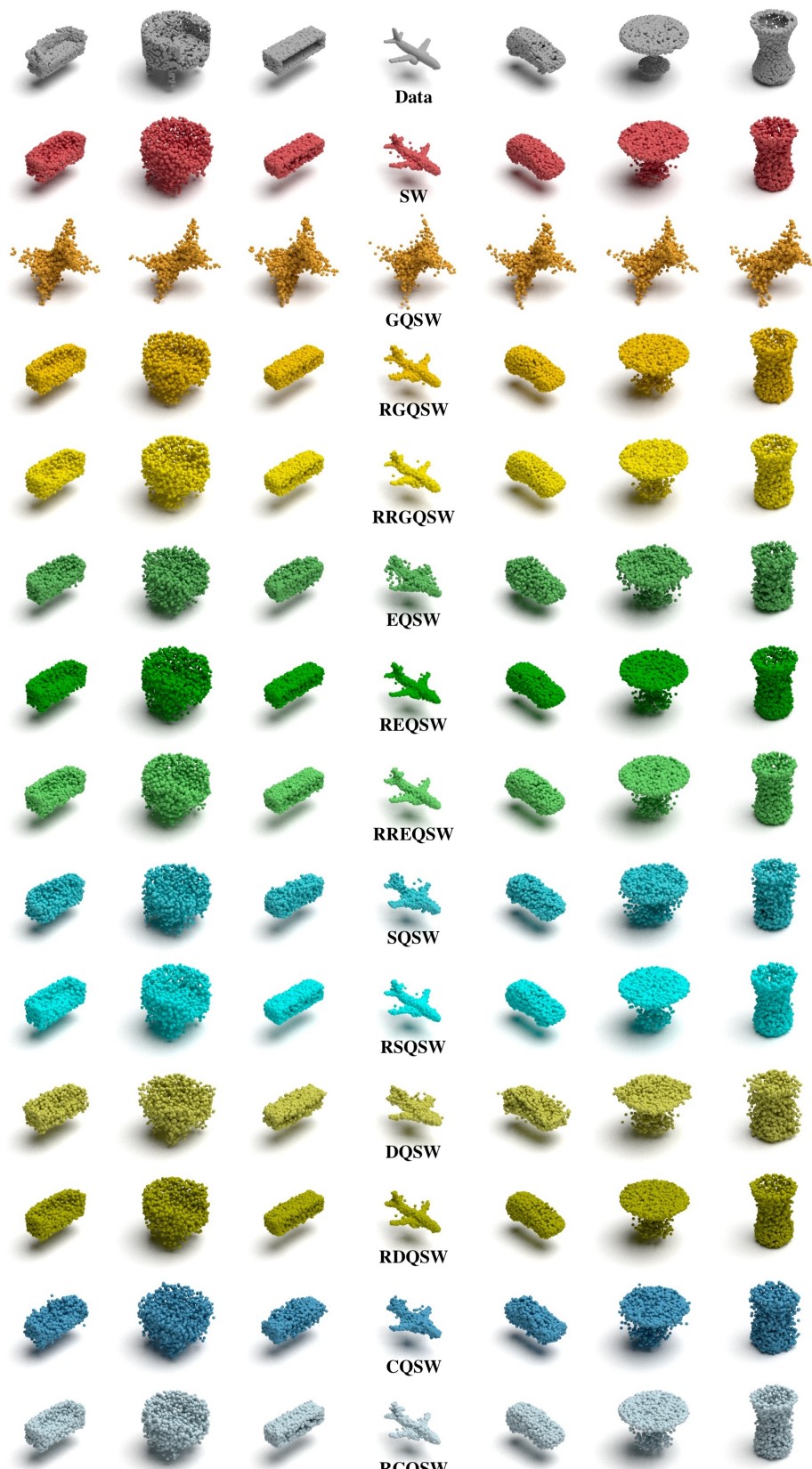

Figure 13: Some reconstructed point-clouds from SW, QSW variants, and RQSW variants with $L = 10$.

Table 6: Reconstruction errors (multiplied by 100) from three different runs of autoencoders trained by different approximations of SW with L=10.

| Distance | Epoch 100 | | Epoch 200 | | Epoch 400 | |
|---|---|---|---|---|---|---|
| | SW$_2$($\downarrow$) | W$_2$($\downarrow$) | SW$_2$ ($\downarrow$) | W$_2$($\downarrow$) | SW$_2$ ($\downarrow$) | W$_2$($\downarrow$) |
| SW L=10 | $2.27 \pm 0.05$ | $10.60 \pm 0.10$ | $2.12 \pm 0.04$ | $9.93 \pm 0.02$ | $1.95 \pm 0.06$ | $9.24 \pm 0.09$ |
| GQSW L=10 | $11.18 \pm 0.06$ | $32.64 \pm 0.06$ | $11.78 \pm 0.07$ | $33.35 \pm 0.07$ | $14.85 \pm 0.03$ | $38.04 \pm 0.04$ |
| EQSW L=10 | $2.53 \pm 0.07$ | $11.82 \pm 0.12$ | $2.29 \pm 0.02$ | $11.03 \pm 0.05$ | $2.11 \pm 0.03$ | $10.40 \pm 0.03$ |
| SQSW L=10 | $2.46 \pm 0.04$ | $11.55 \pm 0.07$ | $2.23 \pm 0.05$ | $10.82 \pm 0.05$ | $2.05 \pm 0.06$ | $10.15 \pm 0.01$ |
| DQSW L=10 | $3.10 \pm 0.03$ | $12.89 \pm 0.04$ | $2.86 \pm 0.07$ | $12.17 \pm 0.10$ | $2.56 \pm 0.03$ | $11.33 \pm 0.08$ |
| CQSW L=10 | $2.60 \pm 0.04$ | $11.92 \pm 0.02$ | $2.44 \pm 0.03$ | $11.29 \pm 0.07$ | $2.25 \pm 0.10$ | $10.59 \pm 0.13$ |
| RGQSW L=10 | $2.27 \pm 0.05$ | $10.60 \pm 0.06$ | $2.10 \pm 0.05$ | $9.92 \pm 0.09$ | $1.95 \pm 0.03$ | $9.20 \pm 0.03$ |
| RRGQSW L=10 | $2.26 \pm 0.02$ | $10.58 \pm 0.03$ | $2.06 \pm 0.08$ | $9.85 \pm 0.12$ | $1.89 \pm 0.06$ | $9.18 \pm 0.07$ |
| REQSW L=10 | $2.26 \pm 0.06$ | $10.57 \pm 0.05$ | $2.09 \pm 0.05$ | $9.91 \pm 0.05$ | $1.91 \pm 0.01$ | $9.20 \pm 0.03$ |
| RREQSW L=10 | $2.24 \pm 0.03$ | $10.54 \pm 0.06$ | $2.06 \pm 0.03$ | $9.85 \pm 0.04$ | $1.88 \pm 0.07$ | $9.17 \pm 0.10$ |
| RSQSW L=10 | $\mathbf{2.23 \pm 0.05}$ | $10.54 \pm 0.08$ | $2.05 \pm 0.03$ | $9.83 \pm 0.03$ | $\mathbf{1.86 \pm 0.03}$ | $9.14 \pm 0.02$ |
| RDQSW L=10 | $2.24 \pm 0.03$ | $\mathbf{10.54 \pm 0.06}$ | $2.05 \pm 0.03$ | $9.85 \pm 0.04$ | $1.87 \pm 0.03$ | $9.14 \pm 0.02$ |
| RCQSW L=10 | $2.24 \pm 0.04$ | $10.55 \pm 0.03$ | $\mathbf{2.03 \pm 0.03}$ | $\mathbf{9.83 \pm 0.03}$ | $1.87 \pm 0.02$ | $\mathbf{9.13 \pm 0.06}$ |

**Recommended variants.** Overall, we recommend RCQSW for this application since it performs well in both settings of $L = 100$ and $L = 10$ in terms of both reconstruction losses and qualitative comparison.

# E  COMPUTATIONAL INFRASTRUCTURE

We use a single NVIDIA V100 GPU to conduct experiments on training deep point-cloud autoencoder. Other applications are done on a desktop with an Intel core I5 CPU chip.

