# OpenReview forum: "Quasi-Monte Carlo for 3D Sliced Wasserstein"
_ICLR.cc/2024/Conference — ICLR 2024 spotlight_

### Official Review · Reviewer_8GjU · 2023-10-26

**Soundness:** 4 excellent
**Presentation:** 4 excellent
**Contribution:** 3 good
**Rating:** 8
**Confidence:** 4

**Summary:**

In this paper, it is proposed to approximate the Sliced-Wasserstein distance (SW) using Quasi-Monte Carlo methods instead of the usual Monte-Carlo approximation. Authors discuss and compare several methods to generate a point set with low spherical cap discrepancy in order to generate uniform samples on the sphere. Then, they also propose a randomized version which is useful to obtain a better estimate of the gradient. Finally, they show the usefulness of QMC approximation of SW on different tasks such as comparison of point clouds, point-cloud interpolation, image style transfer or deep point-cloud autoencoders.

**Strengths:**

Finding better estimates of the Sliced-Wasserstein distance is a nice problem which can improve the results in applications using SW. This work discusses and compares many different methods to construct point sets to generate uniform samples on the sphere, which is not something very common to the best of my knowledge. Furthermore, the results of the application demonstrate well the superiority of the differents methods compared to the usual Monte-Carlo estimator of SW.

- Discussion of different construction of point sets with low discrepancy and allowing to generate uniform samples on the sphere
- Application of these different point sets to approximate SW
- A randomized version to improve the estimate of the gradient
- Different applications which show well the benefits of the different point sets to approximate SW

**Weaknesses:**

- The figures are generally too small which makes them hard to read properly (especially Figure 1 and 2)
- The experiments are only focused on the case $S^2$, which is already nice, but I believe that some of the methods work for higher dimension and could therefore have been tested.

**Questions:**

Why is Proposition 1 not valid for the Maximizing distance version?

"Since the QSW distance does not require resampling the set of projecting directions at each evaluation time, it is faster to compute than the SW distance if QMC point sets have been constructed in advance": I guess we could also use the same samples for the evaluation of different SW (even though I agree this is not really in the spirit of the computation of SW).

---

> ### Author Response · Authors · 2023-11-16
> **Response to Reviewer  8GjU**
>
> We appreciate the time and feedback from the reviewer. We would like to answer the questions as follows:
>
> **Q12**: The figures are generally too small which makes them hard to read properly (especially Figure 1 and 2)
>
> **A12**: Thank you for your comments. We have adjusted the figures to be bigger in the revision.
>
> **Q13**: The experiments are only focused on the case $\mathbb{S}^2$, which is already nice, but I believe that some of the methods work for higher dimension and could therefore have been tested.
>
> **A13**: For an integrand that satisfies some smoothness conditions, we only need $L>2^d$ to make QMC better than MC.  It means that we only need $L>2^3=8$ in 3 dimensions. Although QMC on the unit-hypersphere does not have such a guarantee, it can be conjectured that QMC is still better than MC in relatively low dimensions with a sufficiently large number of samples. Therefore, we focus our study on 2D sphere only in the paper.
>
>
> Some constructions of QMC point-sets can be used in higher dimensions such as Gaussian-based mapping, maximizing distance, and minimizing Coulomb energy. However, in high-dimension, we will need a very large number of points to make QMC better than MC. For example, $d=10$, we need $L > 2^{10} = 1024$ to get better performance. We acknowledge that SW can be used in high-dimensional applications such as deep generative modeling, domain adaptation, and so on. However, we conjecture that the current framework of QMC for SW might not show a clear improvement compared to MC as shown in 3D applications. However, we would like to recall that 3D data plays a crucial role in a lot of applications such as autonomous driving, metaverse, and so on.
>
>
> **Q14**: Why is Proposition 1 not valid for the Maximizing distance version?
>
> **A14**: In Proposition 1, we show that SW is a continuous and bounded function on the unit-hypersphere. To get the asymptotic convergence of the QMC estimation of SW, we need the used QMC pointset to be asymptotic uniform  i.e., when the number of points goes to infinity, the empirical pdf converges to the pdf of uniform distribution. To the best of our knowledge, the point-set created by maximizing distance has not been shown to be asymptotically uniformly distributed. Since this is a problem of numerical analysis, we will leave a careful investigation to future works.
>
> **Q15**: "Since the QSW distance does not require resampling the set of projecting directions at each evaluation time, it is faster to compute than the SW distance if QMC point sets have been constructed in advance": I guess we could also use the same samples for the evaluation of different SW (even though I agree this is not really in the spirit of the computation of SW).
>
> **A15**: Thank you for your insightful comment. We can use a fixed Monte Carlo sample for SW at multiple evaluation times. However, it is equivalent to QMC with a not-good sequence in terms of uniformity as demonstrated in Figure 1 (Figure 5 in the revision). Moreover, we actually do not recommend using deterministic approximation since it is not good for stochastic optimization problems and cannot provide confidence intervals for the population value as mentioned in **Q2** by Reviewer **QEcr**.

---

> > ### Comment · Reviewer_8GjU · 2023-11-22
> >
> > I thank the authors for their response.They address my questions. Thus, I will leave my score unchanged.

---

> > > ### Author Response · Authors · 2023-11-22
> > > **Response to Reviewer 8GjU**
> > >
> > > Thank you again for your insightful feedback.
> > >
> > > Best regards,

---

### Official Review · Reviewer_UGxf · 2023-10-31

**Soundness:** 4 excellent
**Presentation:** 4 excellent
**Contribution:** 3 good
**Rating:** 8
**Confidence:** 3

**Summary:**

This work proposes several novel approximations for the calculation of Sliced Wasserstein distance. Prior works relied on Monte Carlo approximation of expectation in calculating SW distance. The authors propose to substitute it with Quasi-MC approximation of expectation which results in lower approximation error. The main idea behind it is to pick a deterministic set of uniformly distributed points (low-discrepancy sequences) instead of sampling them randomly as in standard MC. The authors propose and evaluate four different strategies to obtain these low-discrepancy sequences on the unit hypersphere, all suitable for QMC calculation of SW distance. They prove that QSW converges to SW in the case of infinite integration points.

Additionally, they propose a randomized version of QSW which they prove to be an unbiased estimator of SW, which allows for unbiased estimation of SW gradients, which leads to improved optimization with RQSW objective.

In the experiments, the authors: (1) show the reduction of the SW approximation error by computing all the different distance approximations between point clouds sampled on the surfaces of 3D shapes; (2) perform point cloud interpolation and image style transfer by interpolation using the gradients of different approximations; (3) train an autoencoder for point clouds using QSW and RQSW as training objectives. (1) and (2) show considerable improvements compared to the baseline SW estimation with MC. For (3) there are improvements but the results seem not to be statistically significant.

**Strengths:**

1. The paper is well-written and is easy to follow.

2. The proposed methods for SW distance approximation seem to consistently outperform the regular MC SW distance approximation.

3. The authors present the approach in a mathematically rigorous manner and prove the main results of the paper.

**Weaknesses:**

1. The improved SW approximation seems not to translate its benefits in large-scale training experiments. It might be worthwhile to recheck that with more powerful auto-encoder architectures, which can better benefit from the improved distance approximation.

2. Figure 1 is not very informative (might be moved to supplementary?), some figures are too small (fig 1, 2), and point clouds in all the figures are too small.

3. There is the complexity analysis of approximations in the paper, but it still will be nice to see a computation time comparison for all the different approximations.

**Questions:**

Related to W.3, empirically, how efficient in terms of time are all the presented approximations?

---
Post rebuttal:
I'd like to thank the reviewers for the additional results and clarifications, most of my concerns were covered. I have increased my rating accordingly.

---

> ### Author Response · Authors · 2023-11-16
> **Response to Reviewer UGxf**
>
> First, we would like to express our gratitude for the time and feedback from the reviewer. We would like to address questions as follows:
>
> **Q8**: The improved SW approximation seems not to translate its benefits in large-scale training experiments. It might be worthwhile to recheck that with more powerful auto-encoder architectures, which can better benefit from the improved distance approximation.
>
>
> **A8**: From Table 2, all QSW variants (except GQSW)  and RQSW variants give lower reconstruction losses (in Wasserstein-2 distance and Slced Wasserstein-2 distance) than SW in all evaluated epochs. This means that the QMC methods help to improve the approximation of training losses which improves the quality of training neural networks. Using a more powerful neural network can lead to a lower reconstruction loss, however, we believe that the comparative order between RQSW and SW might not be changed. When the autoencoder is very powerful, both RQSW and SW will make the reconstruction losses go to 0 when the number of training iterations goes to infinity. However, model misspecification always happens in practice, hence, the reconstruction losses cannot go down further after several training iterations. In the paper, we show that RQSW also helps to reduce the reconstruction losses at early epochs. For QSW, it is deterministic and it must have a large enough number of points to achieve good results. Overall, we still recommend RQSW for applications, especially deep learning applications.
>
> **Q9**: Figure 1 is not very informative (might be moved to supplementary?), some figures are too small (fig 1, 2), and point clouds in all the figures are too small.
>
> **A9**: Thank you for your comments. The main goal of Figure 1 is to demonstrate the low-discrepancy property of point sets. From Figure 1, we can see that the conventional Monte Carlo does not give point-sets with good uniformity. In the revision, we have moved  Figure 1 to the Appendix and scaled up Figure 1 and Figure 2.
>
> **Q10**: There is the complexity analysis of approximations in the paper, but it still will be nice to see a computation time comparison for all the different approximations.
>
> **A10**: Thank you for your suggestion. We have added the computation time to the gradient flow applications in Tables 1-2. For other applications, we omit the computational time since the major computation is not from computing the distance which makes the measurement non-robust to the system noise. However, the comparative comparison should be the same among all applications.
>
> **Q11**: Related to W.3, empirically, how efficient in terms of time are all the presented approximations?
>
>
> **A11**:  From the computational time computed as discussed in Q10, we observe that QSW is always faster than SW since it does not require sampling of projecting directions. RQSW is always slower than QSW since it needs to randomize the constructed QMC point-sets. To compare RQSW and SW, for a relatively small value of the number of points $L$ i.e., 10, SW is slightly faster than RQSW since sampling an orthonormal matrix of size $3\times3$ and applying matrix multiplication costs more time than sampling from the uniform distribution over the unit-hyperspher. However, when $L$ is relatively large i.e., 100, RQSW (with random rotation) is slightly faster than SW. The reason could be sampling $3\times 3$ orthonormal matrix is much faster than sampling $100$ vectors on the unit-hypersphere.

---

### Official Review · Reviewer_Re8P · 2023-11-05

**Soundness:** 2 fair
**Presentation:** 3 good
**Contribution:** 2 fair
**Rating:** 6
**Confidence:** 3

**Summary:**

The paper proposes novel quasi-Monte Carlo (QMC) method for sliced Wasserstein distance (SW), which utilizes various QMC point sets to approximate the integration over uniform distribution on the unit sphere that arises from SW. Specifically, the paper investigates the construction of point sets that approximate the uniform distribution on unit sphere, aiming to obtain lower absolute error, compared to naive MC estimator of SW. Furthermore the paper provides methods to randomize the constructed point sets to obtain unbiasedness, in addition to consistency. Lastly the paper provides empirical study of the error of the proposed estimator, and applications of QMC SW that suggests better performance.

**Strengths:**

The paper is overall clear and well presented, and the results are original and novel to the knowledge of the reviewer. The strengths of the paper includes:
1. The absolute error reduction is a novel perspective, as traditional SW estimators only guarantees consistency and unbiasedness, but a bound for the absolute error is usually missing. This paper sheds new light on the faithfulness of SW estimating beyond MC regime.
2. The paper provides a thorough investigation of the construction of equally-spaced points on the unit sphere, which is of wider interest as it is in general an open problem on 2 dimensions and beyond. The proposed methods all seem promising, and their incorporation into machine learning tasks seems interesting.
3. The paper provides extensive experiments, which seem sufficient for justifying the practicality of the proposed estimators.

**Weaknesses:**

Some weaknesses:
1. The contributions seem limited, as the paper mainly applies existing QMC methods to SW estimation, whereas little was investigated on how QMC and SW interact. Specifically, the paper claims that the Koksma-Hlawka inequality is the main guarantee of lower absolute error. While all listed QMC methods do achieve low discrepancy, on the SW side it does not seem trivial to claim that the SW integrand satisfies the smoothness assumption for the absolute error bound to hold. For instance, for general $\mu,\nu$, the integrand $W_p^p(\theta\mu,\theta\nu)$ only seems to be Lipschitz in $\theta$, which does not imply bounded HK variation in higher dimensions [1]. The BV condition should be verified before claiming the applicability of the inequality.
2. Related to the previous item, when applying the Koksma-Hlawka inequality, the discrepancy was empirical CDF error. It is unclear why the paper then switches to spherical cap discrepancy, and how this is connected to the Koksma-Hlawka inequality. The reviewer agrees that this is a better measurement of the uniformity on sphere, but is not sure about how this directly contributes to bounding the absolute error.

[1] Basu, Kinjal, and Art B. Owen. "Transformations and Hardy--Krause variation." SIAM Journal on Numerical Analysis 54.3 (2016): 1946-1966.

**Questions:**

Please see above (section Weaknesses) for details. Some more questions:
1. The paper used term 'low-discrepancy' twice w.r.t. 2 different discrepancies, and with different benchmark rates $O(L^{-1}\log L ^d)$ and $O(L^{-3/4}\sqrt{\log L})$. How are they related? And which is applicable to obtain the overall error bound for the proposed estimator?
2. Integration over the sphere is a classical numerical analysis problem. Another class of method that seems reasonable is cubature, see [1], with the most notable difference being that the weight is not uniform. Is it possible to use this class of method for estimation of SW? It seems the smoothness requirement for cubature is not any worse than that for QMC in the paper.

[1] Hesse, Kerstin, Ian H. Sloan, and Robert S. Womersley. "Numerical integration on the sphere." Handbook of geomathematics. 2010.

---

> ### Author Response · Authors · 2023-11-16
> **Response to Reviewer Re8P (Part 1)**
>
> First, we would like to thank the reviewer for constructive feedback. We would like to address questions from the reviewer as below:
>
> **Q4**: The contributions seem limited, as the paper mainly applies existing QMC methods to SW estimation, whereas little was investigated on how QMC and SW interact. Specifically, the paper claims that the Koksma-Hlawka inequality is the main guarantee of lower absolute error. While all listed QMC methods do achieve low discrepancy, on the SW side it does not seem trivial to claim that the SW integrand satisfies the smoothness assumption for the absolute error bound to hold. For instance, for general $\mu,\nu$
> , the integrand $W_p^p(\theta\sharp \mu,\theta \sharp \nu)$  only seems to be Lipschitz in  $\theta$, which does not imply bounded HK variation in higher dimensions [1]. The BV condition should be verified before claiming the applicability of the inequality.
>
>
> **A4**: Thank you for pointing this out. We agree that the integrand must satisfy the smoothness assumption for the absolute error bound to hold, and - as far as we currently know - SW is only a Lipchitz function [R1] in the L2 norm. Indeed, we do not claim that the Koksma-Hlawka inequality applies in our setting but use it to suggest that QMC methods might improve SW estimation in practice.  Due to the lack of such theoretical guarantees in our setting, in Section 4.1 we conduct simulations to show that QMC leads to better approximations of SW.
>
> To elaborate further on this point, checking for the needed smoothness conditions of SW as a function of the projecting direction implies working with appropriate notions of derivative on the unit sphere (see, e.g., [R2]).  To the best of our knowledge, the calculation of such derivatives for SW with respect to the projecting direction is highly nontrivial. The hardness of the problem comes from the fact that we do not know the form of the pushforward projected measure and its CDF. Moreover, in the discrete case, the sorting operator is not smooth, hence the calculation is not practical. In addition, SW involves integration on the unit-hypersphere. Therefore, it must be transformed into an integration on the unit cube through a transformation such as the Gaussian-based mapping and equal area mapping, to be considered as a conventional QMC problem.  As discussed in Chapter 6 in [R3], the composition of the transformation and SW must satisfy smoothness assumptions for the error bound to hold. Other related approaches to establishing favorable error rates for QMC designs (e.g., [R4]) rely on showing that the integrand belongs to appropriately defined Sobolev spaces of functions over the unit sphere. This involves working with concepts at the intersection of spherical harmonics and functional analyses that are non-trivial to apply to the SW case, even in simple cases (e.g., when \mu and \nu are both Gaussian) for which closed-form expressions for the slice exist.
> To conclude, we are aware of the importance of deriving theoretical guarantees for QSW estimation and thank the reviewer for further pointing this out. In fact, we are presently considering future research directions to theoretically expand the current work, which nevertheless shows a good practical performance of QMC methods for SW computation.
>
> [R1] Statistical, robustness, and computational guarantees for sliced Wasserstein distances, Nietert et al,
>
> [R2] Spherical harmonics and approximations on the unit sphere: an introduction, Atkinson & Han et al,
>
> [R3] Quasi-Monte Carlo Methods in Non-Cubical Spaces, Basu el al,
>
> [R4] QMC designs: optimal order quasi Monte Carlo integration schemes on the sphere, Brauchart et al,

---

> > ### Author Response · Authors · 2023-11-16
> > **Response to Reviewer Re8P (Part 2)**
> >
> > **Q5**: Related to the previous item, when applying the Koksma-Hlawka inequality, the discrepancy was empirical CDF error. It is unclear why the paper then switches to spherical cap discrepancy, and how this is connected to the Koksma-Hlawka inequality. The reviewer agrees that this is a better measurement of the uniformity on sphere, but is not sure about how this directly contributes to bounding the absolute error.
> >
> >
> > **A5**: Thank you for your insightful question. There are two reasons that spherical-cap discrepancy is used for the QMC problem on the sphere.  Firstly, as in your comment, spherical cap discrepancy is a more natural measure of uniformity on the sphere than the star discrepancy since it considers the geodesic distance on the sphere to compare probability measures.
> >
> > Secondly, as in **Q4**, the error analysis through transformation to use HK inequality leads to a complicated condition-checking problem. Moreover, it is possible to construct point-sets on the unit-hypersphere directly such as generalized spiral points, minimizing Coulomb energy, and maximizing distance. Therefore, the conventional error analysis through HK inequality is not applicable. There is a HK typed inequality for worst-case error with spherical cap discrepancy for functions in reproducing kernel Hilbert space in [R5], however, it is not directly applicable to SW. Moreover, in [R4], for integrand belonging to Hibert spaces with the smoothness parameter $s>d/1$, there exists a point set that leads to a worst-case error of $O(L^{-s/d})$. However, investigating the smoothness order of SW has never been done to the best of our knowledge.
> >
> > Overall, using spherical-cap discrepancy makes the error analysis to be natural and widely applicable for point-sets on the unit-hypersphere. However, the error analysis for SW with spherical-cap discrepancy is still an open question.
> >
> > [R5] Quasi-Monte Carlo rules for numerical integration over the unit sphere S2, Brauchart  et al.
> >
> >
> >
> > **Q6**: The paper used term 'low-discrepancy' twice w.r.t. 2 different discrepancies, and with different benchmark rates
> > $\mathcal{O}(L^{-1}\log L^d)$
> >  and
> > $\mathcal{O}(L^{-3/4}\sqrt{\log L})$
> > . How are they related? And which is applicable to obtain the overall error bound for the proposed estimator?
> >
> >
> > **A6**: There are two notations of low-discrepancy sequences. The first notion is the low-discrepancy sequence on the unit-cube with the star discrepancy at the rate $\mathcal{O}(L^{-1} \log (L)^d$. The second notion is the low-discrepancy sequence on the unit-sphere (2D sphere)  with the spherical cap discrepancy at the second rate. In our paper, we focus on using the spherical cap discrepancy and we verify the low-discrepancy property of discussed point-sets in Figure 1 (Figure 5 in the revision), and Figure 7  in the appendix. From the simulation, we see that there is a positive correlation between the spherical discrepancy and the approximation error for SW.  Again, the theoretical guarantee for the overall error bound for the proposed estimator is still an open question as discussed in previous questions.
> >
> >
> > **Q7**: Integration over the sphere is a classical numerical analysis problem. Another class of method that seems reasonable is cubature, see [1], with the most notable difference being that the weight is not uniform. Is it possible to use this class of method for estimation of SW? It seems the smoothness requirement for cubature is not any worse than that for QMC in the paper.
> >
> >
> > **A7**: Thank you for a very useful reference. The cubature approximation with non-uniform weights average could lead to an interesting class of approximation for SW. For deterministic approximation, we believe it is possible to apply cubature to 3D SW by partitioning the area of the sphere. In this setting,  choosing a point in each partition might play an important role in the performance of the approximation in applications. For example, we could choose the point in each partition in a deterministic fashion or a random fashion. As a result, we might obtain a deterministic approximation or an estimation of SW. Overall, we believe this direction is worth a careful investigation, hence, we leave it in future work.

---

> > > ### Comment · Reviewer_Re8P · 2023-11-20
> > > **Thanks for the response**
> > >
> > > I thank the authors for the response, which addresses all of my questions. I thus leave the rating unchanged.

---

> > > > ### Author Response · Authors · 2023-11-20
> > > > **Response to Reviewer Re8P**
> > > >
> > > > We would like to thank the reviewer again for your insightful feedback.
> > > >
> > > > Please feel free to let us know if you have any further questions during the discussion period!
> > > >
> > > > Best regards,

---

### Official Review · Reviewer_QEcr · 2023-11-05

**Soundness:** 3 good
**Presentation:** 3 good
**Contribution:** 3 good
**Rating:** 8
**Confidence:** 4

**Summary:**

Sliced Wasserstein distance is a commonly used measure of distance between probability measures, whose evaluation involves a usually intractable integral term. Standard evaluation of the integral term is based on Monte Carlo integration, which has an estimation error that depends sub-optimally on the number of samples. This paper fills the hole in literature of computing Sliced Wasserstein distance using Quasi-Monte Carlo methods and Randomized Quasi-Monte Carlo methods. Multiple variants are proposed, and some new ideas for randomization over the sphere are discussed. Adequate experimental results are provided to demonstrate the improvement over Monte Carlo methods.

**Strengths:**

1. The paper is well-written and mostly clear.
2. Adequate background and literature review are provided.
3. The visualizations are very nice.
4. Lots of numerical experiments are conducted and many of them are realistic data examples.

**Weaknesses:**

1. The paper is mostly a combination of existing methods, which lacks certain novelty. However, this paper is a helpful reference for people that needs to numerically compute Sliced Wasserstein distance, so it seems worth publishing in ICLR or somewhere similar.
2. The randomized QMC method for Sliced Wasserstein distance is motivated by stochastic optimization, but it could also be used to obtain confidence intervals on the estimates. This is probably worth discussing, both in theory and with applications.

**Questions:**

In the numerical experiments, while the reported statistics in the tables typically demonstrate significant improvements, such improvement is hard to see from the visualized figures. What are some observable differences in the figures between SW and QSW (e.g. CQSW)?

---

> ### Author Response · Authors · 2023-11-16
> **Response to Reviewer QEcr**
>
> First, we would like to thank the reviewer for the time and feedback. We would like to summarize the questions and answer them as below:
>
> **Q1**: The paper is mostly a combination of existing methods, which lacks certain novelty. However, this paper is a helpful reference for people that needs to numerically compute Sliced Wasserstein distance, so it seems worth publishing in ICLR or somewhere similar.
>
> **A1**: Thank you for your comments. One of the main goals of our paper is to connect the literature on Monte Carlo Methods, especially Quasi-Monte Carlo (QMC) with the sliced Wasserstein literature. We observe that the application of QMC on the unit-hypersphere is quite limited while they are theoretically deep and challenging. In contrast, Sliced Wasserstein has several applications; however, the default computation is carried out via the conventional Monte Carlo integration. Therefore, the paper could be the bridge that makes QMC on the unit-hypersphere be used more in the machine learning and deep learning community and makes the computation of SW more interesting on the theoretical side. In the paper, we demonstrate the favorable performance of the proposal across various applications, e.g., point-cloud interpolation, image-color transfer, and deep point-cloud compression.
>
> We also would like to recall that the randomized Quasi-Monte Carlo on the unit-hypersphere has not been discussed before. In the paper, we introduce two ways to construct a randomized QMC sequence on the unit-hypersphere i.e., mapping a randomized QMC on the unit-cube and random rotating of a QMC sequence on the unit-sphere. We then show that almost all constructions lead to an unbiased estimate of SW and perform well in 3D applications.
>
> **Q2**: The randomized QMC method for Sliced Wasserstein distance is motivated by stochastic optimization, but it could also be used to obtain confidence intervals on the estimates. This is probably worth discussing, both in theory and with applications
>
>
> **A2**: Thank you for your insightful suggestion. The randomized QMC for SW can be seen as a variance reduction technique for estimating SW. In addition, as in your suggestion, confidence intervals on the estimates of SW could also be obtained.
>
> In more detail, we can obtain $M$ i.i.d RQSW estimates, then the corresponding empirical means and empirical variances. Since RQSW is an unbiased estimate of the population SW, we can use the central limit theorem to obtain the asymptotic normality result. After that, we can form the $1-\alpha$ confidence interval for the population value of SW. Alternatively, we can only use Bootstrap to construct a confident interval. We refer the reviewer to the detailed discussion in the revision of our paper, in Appendix B in blue color.
>
> **Q3**: In the numerical experiments, while the reported statistics in the tables typically demonstrate significant improvements, such improvement is hard to see from the visualized figures. What are some observable differences in the figures between SW and QSW.
>
> **A3**: As an example of a visually better performance of QSW and RQSW, we refer the reviewer to Figure 5 (Figure 4 in the revision). In the figure, the reconstructed plane point cloud from SW is quite noisy i.e., some points are far from the main plane. In contrast, the plane from CQSW has only 1 point that is far from the main plane while the plane from RCQSW is not noisy at all. Overall, we agree that it must take a detailed look to see a more favorable performance of QSW and RQSW visually. Therefore, we still recommend using the quantitative result (in Wasserstein-2 distance) as the main comparison between QSW, RQSW, and SW.

---

> > ### Comment · Reviewer_QEcr · 2023-11-20
> >
> > Thank you to the authors for the detailed reply! I have increased my rating accordingly.

---

> > > ### Author Response · Authors · 2023-11-21
> > > **Response to Reviewer QEcr**
> > >
> > > Thank you for raising the score to 8. Please feel free to ask if you have additional questions. We would like to express our gratitude again for your many constructive comments.
> > >
> > > Best regards,

---

### Meta-Review · Area_Chair_RFFW · 2023-12-05

**Metareview:**

The paper proposes a set of methods to estimate the sliced Wasserstein distance (SW) more accurately based on quasi-Monte Carlo (QMC). Several methods to configure QMC points on the unit (2D) sphere are considered, and the paper further extends these methods to allow randomness (while still "more uniform" than MC samples; seemingly holds independent value) to prevent collapse for training models. Convergence/unbiasedness of (most of) the resulting SW estimators are proven. Complexity of resulting algorithms are analyzed. Empirical results have verified improved performance over the conventional MC-based SW estimators. The paper is presented neatly and rigorously.

**Justification For Why Not Higher Score:**

As brought up by Reviewer Re8P, the authors are expected to point out the smoothness assumption for the error bound, and discuss the relation with numerical integration methods on the sphere. The proposed methods, though effective in relatively low dimensions, may not be preferred for high-dimensional problems.

**Justification For Why Not Lower Score:**

All reviewers acknowledged that the paper is well motivated, technically sound, and is conducted and presented with high quality. The recommended method is promising for solving various real 3D point cloud problems.

---

### Decision · Program_Chairs · 2024-01-16

Accept (spotlight)